# Overriding impaired FPR chemotaxis signaling in diabetic neutrophil stimulates infection control in murine diabetic wound

**Ruchi Roy[1,2†], Janet Zayas[1,2,3†], Sunil K Singh[4], Kaylee Delgado[1,2], Stephen J Wood[3], Mohamed F Mohamed[1,2], Dulce M Frausto[1,2], Yasmeen A Albalawi[3], Thea P Price[4], Ricardo Estupinian[1,2], Eileena F Giurini[1,2], Timothy M Kuzel[1,2], Andrew Zloza[1,2], Jochen Reiser[1], Sasha H Shafikhani[1,2,3]***

[1]Department of Medicine, Rush University Medical Center, Chicago, United States; [2]Division of Hematology/Oncology/Cell Therapy, Rush University Medical Center, Chicago, United States; [3]Department of Microbial Pathogens and Immunity, Rush University Medical Center, Chicago, United States; [4]Department of Surgery, Division of Surgical Oncology, University of Illinois at Chicago, Chicago, United States

**\*For correspondence:**
Sasha_Shafikhani@rush.edu

†These authors contributed equally to this work

**Abstract** Infection is a major co-morbidity that contributes to impaired healing in diabetic wounds. Although impairments in diabetic neutrophils have been blamed for this co-morbidity, what causes these impairments and whether they can be overcome, remain largely unclear. Diabetic neutrophils, isolated from diabetic individuals, exhibit chemotaxis impairment but this peculiar functional impairment has been largely ignored because it appears to contradict the clinical findings which blame excessive neutrophil influx as a major impediment to healing in chronic diabetic ulcers. Here, we report that exposure to glucose in diabetic range results in impaired chemotaxis signaling through the formyl peptide receptor (FPR) in neutrophils, culminating in reduced chemotaxis and delayed neutrophil trafficking in the wound of *Lepr*db (db/db) type two diabetic mice, rendering diabetic wound vulnerable to infection. We further show that at least some auxiliary receptors remain functional under diabetic conditions and their engagement by the pro-inflammatory cytokine CCL3, overrides the requirement for FPR signaling and substantially improves infection control by jumpstarting the neutrophil trafficking toward infection, and stimulates healing in diabetic wound. We posit that CCL3 may have therapeutic potential for the treatment of diabetic foot ulcers if it is applied topically after the surgical debridement process which is intended to reset chronic ulcers into acute fresh wounds.

## Editor's evaluation

The data demonstrate substantial neutrophil dysfunction of diabetic or glucose-exposed neutrophils and provide potential therapeutic strategies to improve neutrophil fitness and improve healing of diabetic wounds. The reviewers feel that all their points of concern, suggestions, and comments have been dealt adequately with and that the revised manuscript has improved substantially.

## Introduction

Diabetic foot ulcers are the leading cause of lower extremity amputations in the United States and are responsible for more hospitalizations than any other complication of diabetes (*Sen et al., 2009*; *Brem*

and Tomic-Canic, 2007; Reiber et al., 1999; Frykberg, 2002; Boulton, 2000). Infection with pathogenic bacteria, such as *Pseudomonas aeruginosa*, is a major co-morbidity that contributes to impaired healing in diabetic ulcers (Kirketerp-Møller et al., 2008; Gjødsbøl et al., 2006; Dowd et al., 2008; Redel et al., 2013; Goldufsky et al., 2015). Phagocytic leukocytes, particularly neutrophils (PMNs), play a major role defending wounds from invading pathogens (Martin and Leibovich, 2005). Neutrophils are the first inflammatory leukocytes that infiltrate into the wound (Kim et al., 2008). In addition to their antimicrobial functions mediated by phagocytosis, bursts of reactive oxygen species (ROS), antimicrobial (AMP) production, and neutrophil extracellular trap (NET) (Dovi et al., 2004; Brinkmann et al., 2004), they also express various cytokines and chemokines that set the stage for the subsequent inflammatory and non-inflammatory responses, which further contribute to infection control and partake in healing processes (Velnar et al., 2009; Fenteany et al., 2000; Schäfer and Werner, 2008; Martin, 1997; Diegelmann and Evans, 2004). There appears to be a disconnect in that diabetic ulcers suffer from persistent non-resolving inflammation – characterized by increased neutrophils – yet they fail to control infection. Bactericidal functional impairments in diabetic neutrophils (PMNs) is thought to underlie defective infection control in diabetic wound (Repine et al., 1980; Gallacher et al., 1995). What causes these impairments in diabetic neutrophils remains poorly understood, although the impairment severity has been associated with the degree of hyperglycemia (Repine et al., 1980), suggesting that exposure to high-glucose levels may be a contributing factor to these impairments.

In addition to impaired bactericidal functions, diabetic neutrophils – (isolated from the blood of diabetic patients) – also display impaired chemotactic response (Delamaire et al., 1997). This peculiar functional impairment in diabetic neutrophils has not received much attention primarily because it appears to contradict the clinical findings which finds and blames excessive neutrophil response as a major impediment to healing in chronic diabetic ulcers (Wetzler et al., 2000; Bjarnsholt et al., 2008). Driven by this disconnect and the fact that very little is known about neutrophil trafficking into diabetic wounds particularly early after injury and in response to infection, we sought to assess the possible impact of diabetic neutrophil chemotaxis impairment on the dynamics of neutrophil response and impaired infection control in diabetic wounds.

## Results

### Neutrophil trafficking is delayed in diabetic wounds

We generated full-thickness excisional wounds in *Lepr^{db/db}* (db/db) type two diabetic mice and their normal littermates C57BL/6, as described (Goldufsky et al., 2015; Wood et al., 2014), and challenged these wounds with PA103 *P. aeruginosa* bacteria ($10^3$ CFU/wound), which we have shown to establish a robust and persistent infection and cause wound damage in diabetic mice (Goldufsky et al., 2015). Consistent with our previous report (Goldufsky et al., 2015), db/db wounds contained 2–4 log orders more bacteria than normal wounds, indicating that diabetic wounds are vulnerable to increased infection with *P. aeruginosa* (Figure 1—figure supplement 1). We next collected wound tissues on days 1, 3, 6, and 10 post-infection and assessed them for their neutrophil contents by immunohistochemistry (IHC) using the neutrophil marker Ly6G (Wong et al., 2015; Kroin et al., 2016). Surprisingly, diabetic mice exhibited substantially reduced neutrophil influx in wounds early after injury at days 1 and 3 but significantly higher neutrophil contents in day 6 and day 10, as compared to normal wounds (Figure 1a–b). Corroborating these data, myeloperoxidase (MPO) -– (a marker for primarily activated neutrophils Klebanoff, 2005) – was also substantially reduced in diabetic wounds early after injury at days 1 and 3 but significantly higher in day 10 wounds (Figure 1c). Assessment of neutrophil contents in day one normal and diabetic infected wounds by flow cytometry – where neutrophils were identified as CD45^+Ly6C/G^hiCD11b^hi (Kuijpers et al., 1991; Atzeni et al., 2002) – further corroborated the inadequate neutrophil trafficking into diabetic wounds early after injury (Figure 1d and Figure 1—figure supplement 2). These data indicated that neutrophil response – (which is needed to combat infection) – is delayed in diabetic wounds, rendering these wounds vulnerable to infection early after injury.

### Chemotactic response through the FPR is impaired in diabetic neutrophils

Depending on the tissue or the condition, neutrophil trafficking in response to injury and/or infection occurs in multiple waves mediated by ~30 receptors on neutrophils and involves multiple signaling

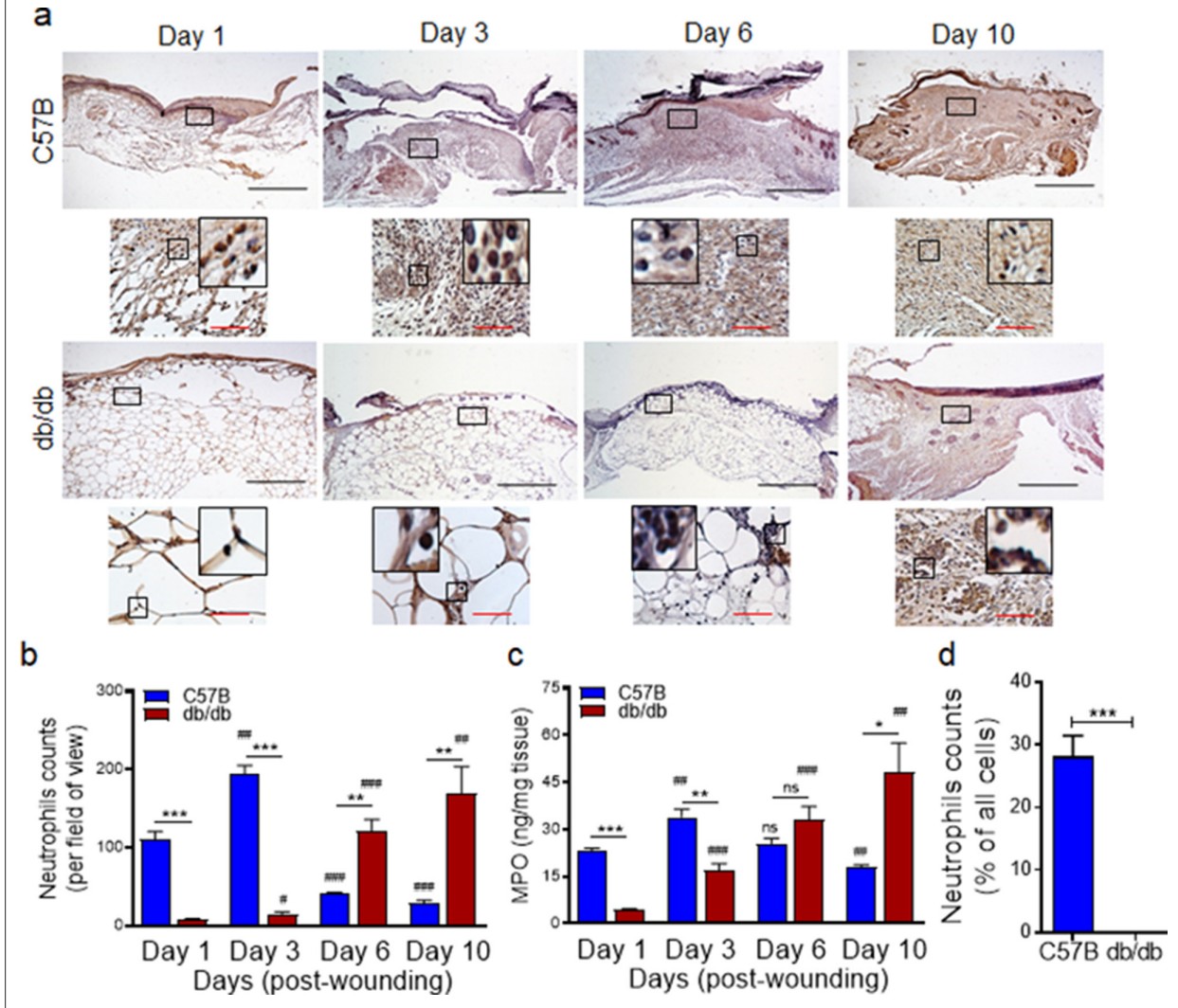

**Figure 1.** Neutrophil response is delayed in infected diabetic wound tissue. Normal (C57BL/6) and diabetic (db/db) wounds were infected with PA103 (1000 CFU/wound). (**a–b**) Wound tissues were harvested at indicated timepoints post-infection and assessed for neutrophil contents by histological analysis using anti-Ly6G antibody. (**a**) Representative regions from underneath the wounds extending in the dermis are shown at ×40 and ×400 magnification (top and bottom, respectively). A representative magnified region is also inserted in the ×400 magnification images. Black scale bar = 500 μm for ×40 magnification and red scale bar = 50 μm for ×400 magnification. (**b**) The corresponding data were plotted as the Mean ± SEM. (**c**) Wounds at indicated timepoints were assessed for their MPO contents by ELISA and the tabulated data are shown as the Mean ± SEM. (**d**) Day 1 infected wound tissues of C57BL/6 and db/db were evaluated for their neutrophil contents by flow cytometry. Corresponding data were plotted as the Mean ± SEM. (N = 4; ns = not significant, *p < 0.05; **p < 0.01; ***p < 0.001 – are comparisons made between C57BL/6 and db/db at indicated timepoints; or #p < 0.05; ##p < 0.01; ###p < 0.001 are comparisons made within each group to day one values, respectively. Statistical analyses between groups were conducted by One-way ANOVA with additional post hoc testing, and pair-wise comparisons between groups were performed or by unpaired Student's *t*-test).

The online version of this article includes the following source data and figure supplement(s) for figure 1:

**Source data 1.** Related to *Figure 1b*.

**Source data 2.** Related to *Figure 1c*.

**Source data 3.** Related to *Figure 1d*.

**Figure supplement 1.** Diabetic wound is vulnerable to increased infection with *Pseudomonas aeruginosa*.

**Figure supplement 1—source data 1.** Related to *Figure 1—figure supplement 1*.

**Figure supplement 2.** Gating strategy for flow cytometric analysis.

pathways (*de Oliveira et al., 2016*; *Liu et al., 2012*; *Sadik et al., 2011*; *Futosi et al., 2013*; *Ng et al., 2011*; *Afonso et al., 2012*; *Chou et al., 2010*). However, the initial neutrophil chemotaxis in response to injury or infection involves the activation of G protein–coupled formyl peptide receptor (FPR) by *N*-formyl peptides, such as fMet-Leu-Phe (fMLF, a.k.a., fMLP), which is released either by injured tissues or by invading bacteria (*de Oliveira et al., 2016*; *Roupé et al., 2010*). FPR1 and FPR2 are two FPR implicated in these responses, although FPR1 appears to be the primary FPR in responding to infection, as it has significantly higher affinity for bacterial formyl peptides, whereas FPR2 has a broader range of ligands than FPR1 and has been implicated in the resolution of inflammation in response to pro-resolving agonists, such as Annexin A1 (*Ye et al., 2009*; *Jeong and Bae, 2020*; *Bena et al., 2012*; *Yazid et al., 2012*; *Serhan and Savill, 2005*). Activation of FPR then leads to the upregulation and secretion of lipid signals, such as the leukotriene B$_4$ (LTB$_4$), which in turn activate BLT1, (another G-protein-coupled receptor on neutrophils), amplifying neutrophil trafficking by enhancing the signaling through FPR (*Afonso et al., 2012*). BLT1 activation in neutrophils by LTB$_4$ also results in upregulation and secretion of pro-inflammatory cytokines, particularly IL-1β which in turn induces the expression and secretion of other ligands (i.e. CCL3 and CXCL1) in tissue resident epithelial cells and inflammatory leukocytes, which further amplify neutrophil trafficking and other inflammatory leukocytes including monocytes, by engaging their respective auxiliary receptors, such as CCR1 and CXCR2 (*36, 37, 44, 45*).

To assess the role of chemotaxis impairment in reduced neutrophil influx into diabetic wounds early after injury, we isolated neutrophils from the blood of normal and diabetic mice and assessed chemotaxis signaling through FPR in response to fMLF. Compared to normal neutrophils isolated from C57BL/6, db/db neutrophils were significantly impaired in their ability to chemotax toward fMLF (*Figure 2a*). Consistent with reduced signaling through the FPR in response to bacterial fMLF, expression of FPR1 was significantly diminished in db/db neutrophils, as assessed by western blotting (*Figure 2b–c*). Further corroborating these data, the percentage of FPR1-positive neutrophils was significantly reduced in day one diabetic wounds, after accounting for reduced number of neutrophils in diabetic wounds early after injury by assessing equal number of neutrophils by flow cytometry (*Figure 2d*).

Various studies have shown direct correlations between plasma glucose levels and prevalence and/ or severity of infection in diabetic patients (*Rayfield et al., 1982*; *Latham et al., 2001*; *Zerr et al., 1997*), suggesting that exposure to high glucose levels may be responsible for impaired neutrophil functions in diabetes. Consistent with these reports, short-term and long-term glycemic control in diabetic rats, has been shown to significantly improve their ability to control *Staphylococcus aureus* infection (*Kroin et al., 2015*). To assess the impact of high glucose on signaling through the FPR, we purified neutrophils from human blood and C57BL/6 mice bone marrow (*Figure 2—figure supplement 1* and Materials and methods), incubated them in media containing glucose in the normal range (90 mg/dl) or in the diabetic range (200–500 mg/dl) for 1 hr, and evaluated their chemotactic responses toward fMLF. Of note, 1 hr exposure to high glucose in diabetic range had no effect on viability of neutrophils.

Exposure to high glucose levels caused significant reduction in chemotactic response to fMLF in both human and mouse neutrophils (*Figure 2e–f*). While neutrophils exposed to normal glucose showed a bell-shaped curve in their chemotaxis response toward fMLF concentrations (0.01–1000 nM) with 100 nM being the optimum concentration, neutrophils exposed to high glucose showed flat chemotaxis response toward these fMLF concentrations, trending toward lower chemotaxis at higher concentrations (*Figure 2—figure supplement 1*), indicating that high fMLF ligand concentrations cannot rescue chemotaxis signaling through FPR in neutrophils exposed to high glucose. The bell-shaped response to fMLF in normal neutrophils is in line with previous reports showing reduction in neutrophil chemotactic responses to other ligands at high concentrations (*Gomez-Cambronero et al., 2003*; *Burnett et al., 2017*). Of note, exposure to high glucose also caused significant reductions in FPR1 surface expression, FPR1 and PLCγ protein levels, as well as cAMP levels (*Figure 2g–k*), which are all required to mediate FPR-mediated chemotaxis in neutrophils (*Afonso et al., 2012*; *Heit et al., 2002*; *Hirsch et al., 2000*). Corroborating these data, 1 hr exposure to high glucose resulted in significant reductions in the *FPR1* and *PLCγ*transcription as determined by mRNA analysis by RT-PCR (*Figure 2l–m*).

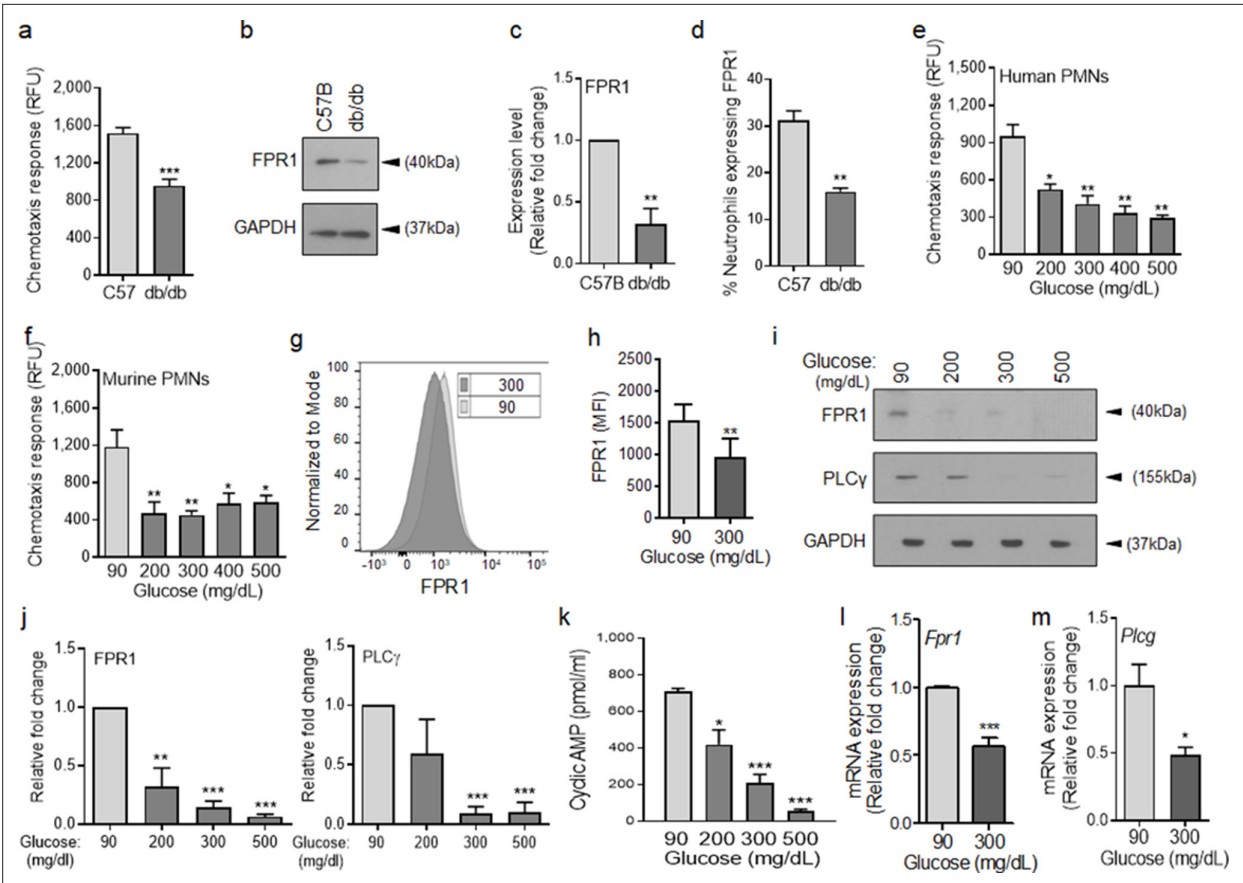

**Figure 2.** Chemotactic response is impaired in diabetic neutrophils through FPR. (**a–b**) Neutrophils were isolated from the peripheral blood of C57BL/6 and db/db animals to assess: (**a**) their ability to chemotax toward 100 nM fMLP, or (**b**) for the expression of FPR1 by Western blotting. (**c**) Densitometry values associated with (**b**) are plotted as Mean ± SEM (N = 4 blood pools/group, each blood pool was from 4 mice). (**d**) Equal number of neutrophils (isolated from Day 1 C57B and db/db wounds) were assessed for the surface expression of FPR1 on neutrophils by flow cytometry (N = 3 mice/group). (**e–f**) Purified neutrophils from peripheral blood of non-diabetic individuals (**e**), or C57BL/6 bone marrow (**f**), were exposed to media containing glucose in normal range (90 mg/dl) or in diabetic range (200–500 mg/dl) for 1 hr to assess their ability to chemotax toward 100 nM fMLP. Data are plotted as Mean ± SEM. (N > 4). (**g–h**) Neutrophils from C57BL/6 bone marrow were exposed to glucose in normal range (90 mg/dl) or in diabetic range (300 mg/dl) for 1 hr and assessed for surface expression of FPR1 by flow cytometry. A representative histogram is shown in (**g**) and the corresponding tabulated data, plotted as Mean ± SEM is shown in (**h**) (N = 3). (**i–j**) Murine neutrophils (from C57B bone marrow) were exposed to glucose in normal or diabetic range (90 mg/dl or 300 mg/dl) for 1 hr and assessed for the expression of indicated proteins by Western blotting. Representative Western blots are shown in (**i**) and corresponding densitometry values, plotted as Mean ± SEM, are shown in (**j**). (N ≥ 3 independent experiments). (**k–m**) Murine neutrophils exposed to normal or diabetic glucose, as described for (**g–h**), were assessed for Cyclic AMP production by ELISA (**k**), and for mRNA of Fpr1 and Plcγ by RT-PCR (**l-m**). (N ≥ 3, ns = not significant, *p < 0.05, **p < 0.01, ***p < 0.001. Statistical analyses between groups were conducted by One-way ANOVA with additional post hoc testing, and pair-wise comparisons between groups were performed or by unpaired Student's *t*-test).

The online version of this article includes the following source data and figure supplement(s) for figure 2:

**Source data 1.** Related to *Figure 2a*.

**Source data 2.** Related to *Figure 2b*.

**Source data 3.** Related to *Figure 2c*.

**Source data 4.** Related to *Figure 2d*.

**Source data 5.** Related to *Figure 2e*.

**Source data 6.** Related to *Figure 2f*.

**Source data 7.** Related to *Figure 2h*.

**Source data 8.** Related to *Figure 2i*.

**Source data 9.** Related to *Figure 2j*.

**Source data 10.** Related to *Figure 2k*.

*Figure 2 continued on next page*

*Figure 2 continued*

**Source data 11.** Related to *Figure 2l*.
**Source data 12.** Related to *Figure 2m*.

---

**Figure supplement 1.** Chemotactic response is impaired in diabetic neutrophils through FPR.
**Figure supplement 1—source data 1.** Related to *Figure 2—figure supplement 1d*.
**Figure supplement 2.** Exposure to high glucose dampens the expression of FPR1 in neutrophils.
**Figure supplement 2—source data 1.** Related to *Figure 2—figure supplement 2a*.
**Figure supplement 2—source data 2.** Related to *Figure 2—figure supplement 2b*.
**Figure supplement 2—source data 3.** Related to *Figure 2—figure supplement 2c*.
**Figure supplement 3.** Exposure to high glucose dampens the expression of FPR2 in neutrophils.
**Figure supplement 3—source data 1.** Related to *Figure 2—figure supplement 3a*.
**Figure supplement 3—source data 2.** Related to *Figure 2—figure supplement 3b*.
**Figure supplement 3—source data 3.** Related to *Figure 2—figure supplement 3c*.

---

To assess whether the adverse impact of high glucose on FPR1 expression was transient or sustained, we exposed purified neutrophils to glucose at 90 or 300 mg/dl and assessed the expression of FPR1 by RT-PCR and by western blotting after 1, 2, or 3 hr post exposure. Data indicated that exposure to high glucose significantly reduced the expression of FPR1 both at the transcriptional and translational levels at all timepoints, indicating that exposure to high glucose dampens the expression of FPR1 in a sustained manner (*Figure 2—figure supplement 2*). Of note, high glucose similarly dampened the expression of FPR2 both at transcriptional and translational levels, indicating that the adverse impact of high glucose is not restricted to FPR1 (*Figure 2—figure supplement 3*). Collectively, these data indicated that elevated glucose levels in diabetes is responsible for the reduced chemotactic response through FPR in diabetic neutrophils.

## Some auxiliary chemotaxis receptors remain functional under diabetic conditions

Although, the initial neutrophil chemotactic response through FPR and the amplification of neutrophil chemotactic responses via other auxiliary receptors are interconnected and occur sequentially in vivo (*Liu et al., 2012*; *Sadik et al., 2011*; *Futosi et al., 2013*; *Ng et al., 2011*; *Afonso et al., 2012*; *Chou et al., 2010*), none of these receptors appear to be essential on their own and their defects can be overcome by engaging other receptors (*Chou et al., 2010*; *Lämmermann et al., 2013*; *Park et al., 2009*). Chronic diabetic ulcers suffer from increased neutrophil contents (*Wetzler et al., 2000*; *Bjarnsholt et al., 2008*), indicating that diabetic neutrophils are capable of migrating into the wound, albeit at dysregulated kinetics as our data show (*Figure 1*). Together, these findings suggested that chemotactic responses of diabetic neutrophils – although impaired through the FPR (*Figure 2* and *Figure 2—figure supplement 1*) – may be functional through one or more auxiliary receptors that mediate the amplification phase of neutrophil trafficking in wound and toward infection.

To evaluate this possibility, we assessed chemotactic responses toward CCL3 in human and mouse neutrophils after 1 hr exposure to glucose at normal or diabetic levels. The reason we focused on CCL3 was because it engages multiple auxiliary receptors, namely CCR1, CCR4, and CCR5 (*Ramos et al., 2005*; *da Silva et al., 2017*; *Yoshie and Matsushima, 2015*). Of note, CCR1 is an important receptor that is implicated in neutrophil trafficking to post-ischemic tissues (*Reichel et al., 2006*) and ischemia is an important co-morbidity associated with impaired healing in diabetic wound (*Brem and Tomic-Canic, 2007*; *Armstrong et al., 1998*). Data indicated that exposure to glucose in the diabetic range did not affect the chemotactic responses toward CCL3 in human or mouse neutrophils (*Figure 3a–b*), suggesting that these auxiliary receptors are unaffected by high glucose. To corroborate these data, we assessed the impact of high glucose exposure on CCR1 auxiliary receptor. In line with chemotaxis data, CCR1 expression remained unaffected in neutrophils after exposure to high glucose for 1 hr as assessed by Western blotting (*Figure 3c–d*), by mRNA analysis (*Figure 3e*), and by surface expression analysis (*Figure 3f–g*). Further corroborating these data, CCR1 expression was similar in neutrophils isolated from the blood of db/db and C57BL/6 mice (*Figure 3h–i*), and

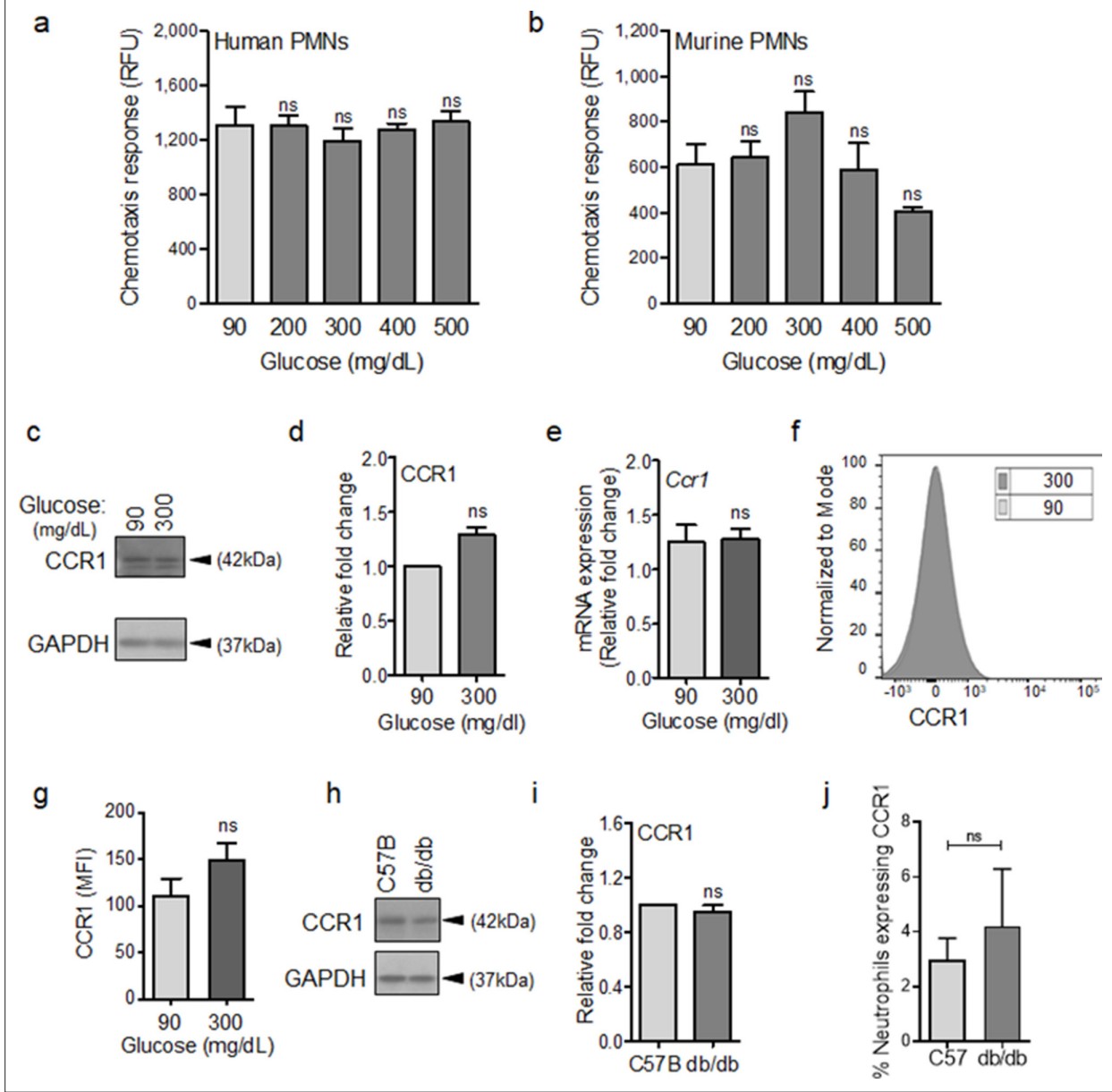

**Figure 3.** CCR1 receptor remains functional under diabetic conditions. Human (**a**) or mouse (**b**) neutrophils were examined for their chemotactic responses toward CCL3 (5 ng/ml) after 1 hr exposure to glucose in normal (90 mg/dl) or diabetic range (200–500 mg/dl). (N > 3). (**c–e**) Neutrophils isolated from bone marrow of C57BL/6 were exposed to normal glucose (90 mg/dl) or high glucose (300 mg/dl) for 1 hr and assessed for CCR1 expression by western blotting (**c–d**) and for mRNA transcription analysis by RT-PCR. (N = 5 for western blots and N = 4 for RT-PCR). (**f–g**) Neutrophils isolated from bone marrow of C57BL/6 were exposed to normal glucose (90 mg/dl) or high glucose (300 mg/dl) for 1 hr and assessed for CCR1 surface expression by flow cytometry. A representative histogram is shown in (**f**) and the corresponding data, plotted as Mean ± SEM, is shown in (**g**) (N = 4). (**h–i**) Neutrophils isolated from peripheral blood of db/db and C57BL/6 mice were assessed for the expression of CCR1 by western blotting. A representative western blot is shown in (**h**) and the corresponding tabulated values are shown in (**i**). (N = 4 mice/group). (**j**) Equal numbers of neutrophils from day 1 C57BL/6 and db/db infected wounds were assessed for CCR1 surface expression by flow cytometry. (N = 3). (Statistical analyses between groups were conducted by One-way ANOVA with additional post hoc testing, and pair-wise comparisons between groups were performed or by unpaired Student's *t*-test; ns = not significant, *$p < 0.05$, **$p < 0.01$, ***$p < 0.001$).

The online version of this article includes the following source data and figure supplement(s) for figure 3:

**Source data 1.** Related to *Figure 3a*.

**Source data 2.** Related to *Figure 3b*.

**Source data 3.** Related to *Figure 3c*.

*Figure 3 continued on next page*

*Figure 3 continued*

**Source data 4.** Related to *Figure 3d*.

**Source data 5.** Related to *Figure 3e*.

**Source data 6.** Related to *Figure 3g*.

**Source data 7.** Related to *Figure 3h*.

**Source data 8.** Related to *Figure 3i*.

**Source data 9.** Related to *Figure 3j*.

**Figure supplement 1.** Exposure to high glucose does not affect CXCR2 auxiliary receptor.

**Figure supplement 1—source data 1.** Related to *Figure 3—figure supplement 1b*.

**Figure supplement 1—source data 2.** Related to *Figure 3—figure supplement 1c*.

the percentage of CCR1-positive neutrophils in db/db day 1 wounds were similar to C57BL/6 day 1 wounds, after accounting for the reduced number of leukocytes in day one diabetic wounds by assessing equal number of neutrophils by flow cytometry (*Figure 3j*). Of note, surface expression of auxiliary receptor CXCR2, (another important auxiliary receptor involved in the amplification of neutrophil response in wound and toward infection [*de Oliveira et al., 2016*; *Brubaker et al., 2013*]), on neutrophils and chemotaxis through the CXCR2 in response to CXCL1 (a.k.a. KC) – a known ligand for CXCR2 (*Chintakuntlawar and Chodosh, 2009*) – were also unaffected by high glucose exposure in neutrophils (*Figure 3—figure supplement 1a-c*). Collectively, these data suggested that at least CCR1 and CXCR2 auxiliary receptors may remain functional under diabetic conditions.

## Topical treatment with CCL3 bypasses the requirement for FPR signaling and enhances neutrophil trafficking and infection control in diabetic wound

If auxiliary receptors seem to be unaffected under diabetic conditions as our data in *Figure 3* and *Figure 3—figure supplement 1* indicate, why is neutrophil trafficking so severely diminished in diabetic wounds early after injury (*Figure 1*). As discussed above, production of ligands (including CCL3) for auxiliary receptors in tissue ultimately depends on FPR activation (*Afonso et al., 2012*; *Chou et al., 2010*; *Su and Richmond, 2015*; *Luster et al., 2005*). In addition, leukocytes (i.e. neutrophils) are major cellular sources of ligands for auxiliary receptors (including CCL3) (*Ridiandries et al., 2018*; *Tecchio et al., 2014*; *Sanz and Kubes, 2012*). Therefore, reduced neutrophil response in diabetic wounds early after injury (*Figure 1*) could also adversely affect the production of ligands for auxiliary receptors in diabetic wounds early after injury (including CCL3). Moreover, increased expression of immunosuppressive IL-10 in diabetic wounds early after injury has been shown to lead to significant reduction in toll-like receptor (TLR) signaling in diabetic wounds early after injury (*Roy et al., 2021*). And TLR signaling has been implicated in the production of ligands (including CCL3) for auxiliary receptors (*Kochumon et al., 2020*; *Ahmad et al., 2019*). Taken all these into account, we posited that although auxiliary receptors on neutrophils may remain functional under diabetic condition, they may not be functioning in diabetic wounds early after injury because of inadequate expression of their ligands. We assessed the expression of CCL3 in day one normal and diabetic wounds infected with *P. aeruginosa*. In line with our hypothesis, CCL3 expression was substantially reduced in day one diabetic wounds, as assessed by mRNA analysis and Western blotting (*Figure 4a–c*). These data suggested that although auxiliary receptors on neutrophils may remain functional under diabetic condition, they may not be functioning to recruit neutrophils in diabetic wounds early after injury because of inadequate ligands' production for the auxiliary receptors. If this is the case, augmenting diabetic wounds with CCL3 early after injury should be able to override deficiency in the FPR signaling and enhance neutrophil migration into diabetic wounds.

To test our hypothesis, we treated db/db wounds topically with CCL3 (1 µg/wound) prior to infection and assessed its impact on neutrophil response and infection control in diabetic wounds. Consistent with our hypothesis, one-time topical treatment with CCL3 significantly increased neutrophil trafficking in day one diabetic wounds, as assessed by Ly6G histological analysis (*Figure 4d–e*), by flow cytometry (*Figure 4f*), and by MPO analysis (*Figure 4g*). Importantly, CCL3 treatment significantly enhanced the ability of diabetic wounds to control infection, as demonstrated by nearly a

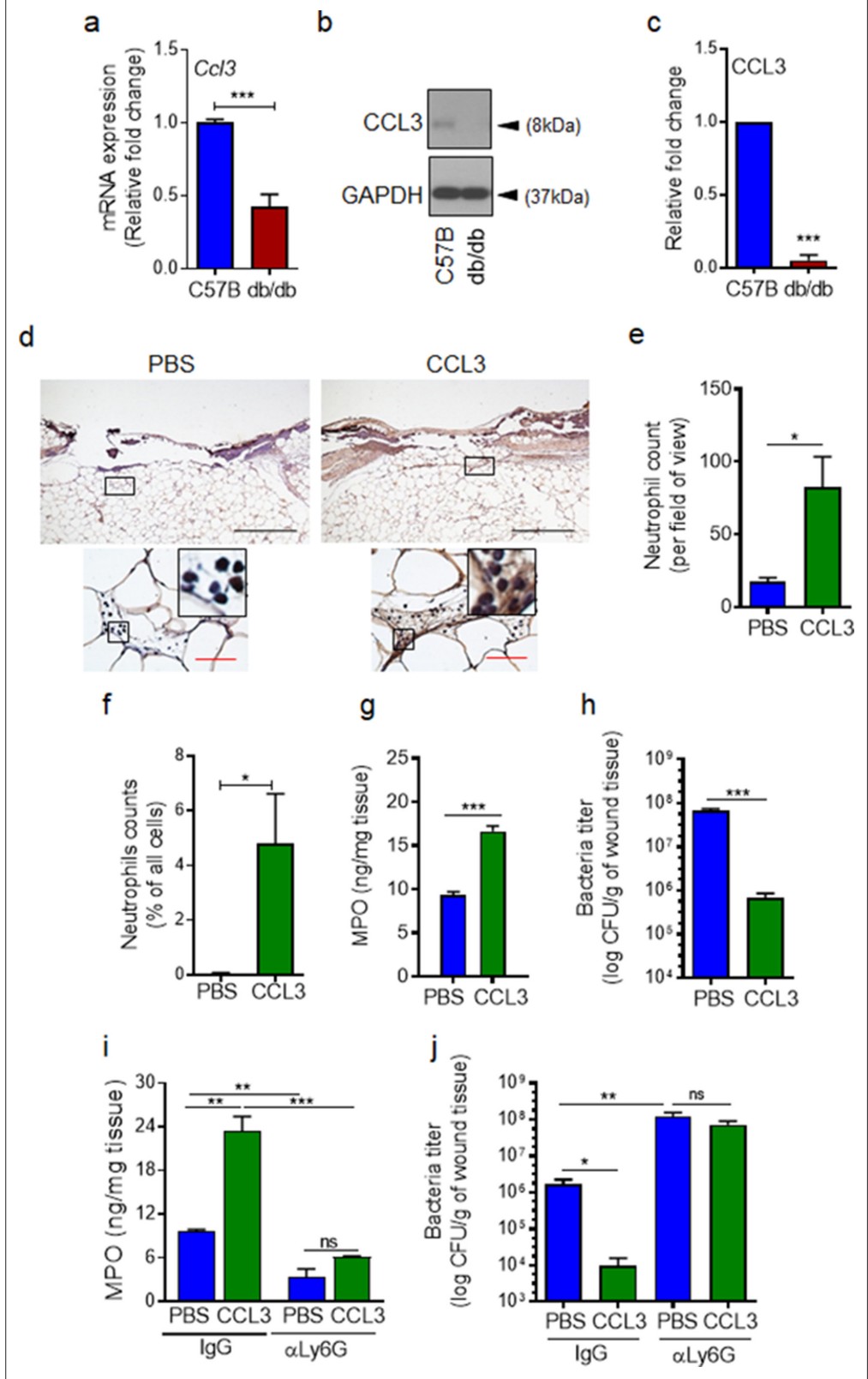

**Figure 4.** CCL3 topical treatment enhances neutrophil response and infection control in diabetic wound. (**a–c**) Day 1 wound tissues of C57BL/6 and db/db infected wounds were harvested and assessed for the CCL3 mRNA levels by RT-PCR (**a**) and by western blotting (**b–c**), and the data were plotted as the Mean ± SEM, after normalization to 18 S and GAPDH, respectively (N = 6 mice/group for (**a**) and 4 mice/group for (**b–c**)). (**d-e**) db/db diabetic

*Figure 4 continued on next page*

*Figure 4 continued*

wounds were treated with either PBS or CCL3 (1 µg/wound) and infected with PA103 (1000 CFU/wound). Twenty-four h post-infection, wounds were collected and assessed for their neutrophil contents by histological analysis using anti-Ly6G antibody. (**d**) Representative wound images at ×40 and ×400 magnification (top and bottom, respectively) are shown. Inserts are representative magnified regions within the ×400 magnification images. (Black scale bar = 500 µm for ×40 magnification and red scale bar = 50 µm for ×400 magnification). (**e**) Corresponding data associated with (**d**) are plotted as Mean ± SEM. (N = 4 mice/group) (**f**) Neutrophil contents of PBS or CCL3-treated db/db infected wounds at day 1 were assessed by flow cytometry (**f**) or by MPO analysis (**g**) and the data were plotted as Mean ± SEM. (N > 3 mice/group for (**f**) and N = 4 mice/group for (**g**)). (**h–i**) db/db mice received either α-Ly6G (100 µg/mouse) to cause neutrophil depletion or α-IgG isoform as control, by intraperitoneal (i.p.) injection. Twenty-four hr after injection, α-IgG or α-Ly6G-treated animals were wounded and treated with either PBS or CCL3 and infected with PA103. The impact of neutrophil depletion on the ability of CCL3 treatment to boost infection control in diabetic wound was assessed by MPO analysis (**i**) and CFU count determination (h & j) in day 1 wounds. Data were plotted as Mean ± SEM. (N = 4 mice/group for (**h**); N = 3 mice/group for (**i**); and N > 4 mice/group for (**j**). ns = not significant, *p < 0.05; **p < 0.01, ***p < 0.001. Statistical analyses between groups were conducted by One-way ANOVA with additional post hoc testing, and pair-wise comparisons between groups were performed or by unpaired Student's *t*-test.).

The online version of this article includes the following source data and figure supplement(s) for figure 4:

**Source data 1.** Related to *Figure 4a*.

**Source data 2.** Related to *Figure 4b*.

**Source data 3.** Related to *Figure 4c*.

**Source data 4.** Related to *Figure 4e*.

**Source data 5.** Related to *Figure 4f*.

**Source data 6.** Related to *Figure 4g*.

**Source data 7.** Related to *Figure 4h*.

**Source data 8.** Related to *Figure 4i*.

**Source data 9.** Related to *Figure 4j*.

**Figure supplement 1.** Supplementary data associated with *Figure 4*.

**Figure supplement 1—source data 1.** Related to *Figure 4—figure supplement 1b*.

two log-order reduction in the number of bacteria contained in the CCL3-treated db/db wounds (*Figure 4h*).

To assess the dependence enhanced infection control on neutrophils in CCL3-treated diabetic wounds, we depleted db/db mice of neutrophils by anti-Ly6G antibody (*Nozawa et al., 2006*), 24 hr prior to wounding and assessed the impact of neutrophil depletion on the ability of CCL3-treated db/db wounds to control *P. aeruginosa* infection. Anti-Ly6G reduced the neutrophil contents in circulation by ~97% and in wound by ~75% (*Figure 4i* and *Figure 4—figure supplement 1a,b*). Neutrophil-depletion also resulted in ~2 log-order more bacteria in diabetic wounds, indicating that despite their known bactericidal functional impairments (*Repine et al., 1980*; *Gallacher et al., 1995*), diabetic neutrophils still contribute to a degree in infection control in these wounds (*Figure 4j*). Importantly, neutrophil-depletion abrogated CCL3's beneficial effects in boosting antimicrobial defenses against *P. aeruginosa* in diabetic wounds (*Figure 4j*), indicating that CCL3-induced enhanced infection control in diabetic wound is dependent on its ability to enhance neutrophil response in diabetic wound.

## Treatment with CCL3 does not lead to persistent non-resolving inflammation in infected diabetic wounds and stimulates healing

Although, treatment with CCL3 substantially improved diabetic wound's ability to control infection by enhancing neutrophil response in day one wounds (*Figure 4*), it remained a possibility that CCL3 treatment could have long-term adverse consequences, as it could lead to heightened inflammatory environment which would be detrimental to the process of tissue repair and healing in diabetic wounds. Afterall, persistent non-resolving inflammation, (as manifested by increases in pro-inflammatory cytokines and neutrophils), is considered a major contributor to healing impairment in diabetic foot ulcers (*Wetzler et al., 2000*; *Bjarnsholt et al., 2008*).

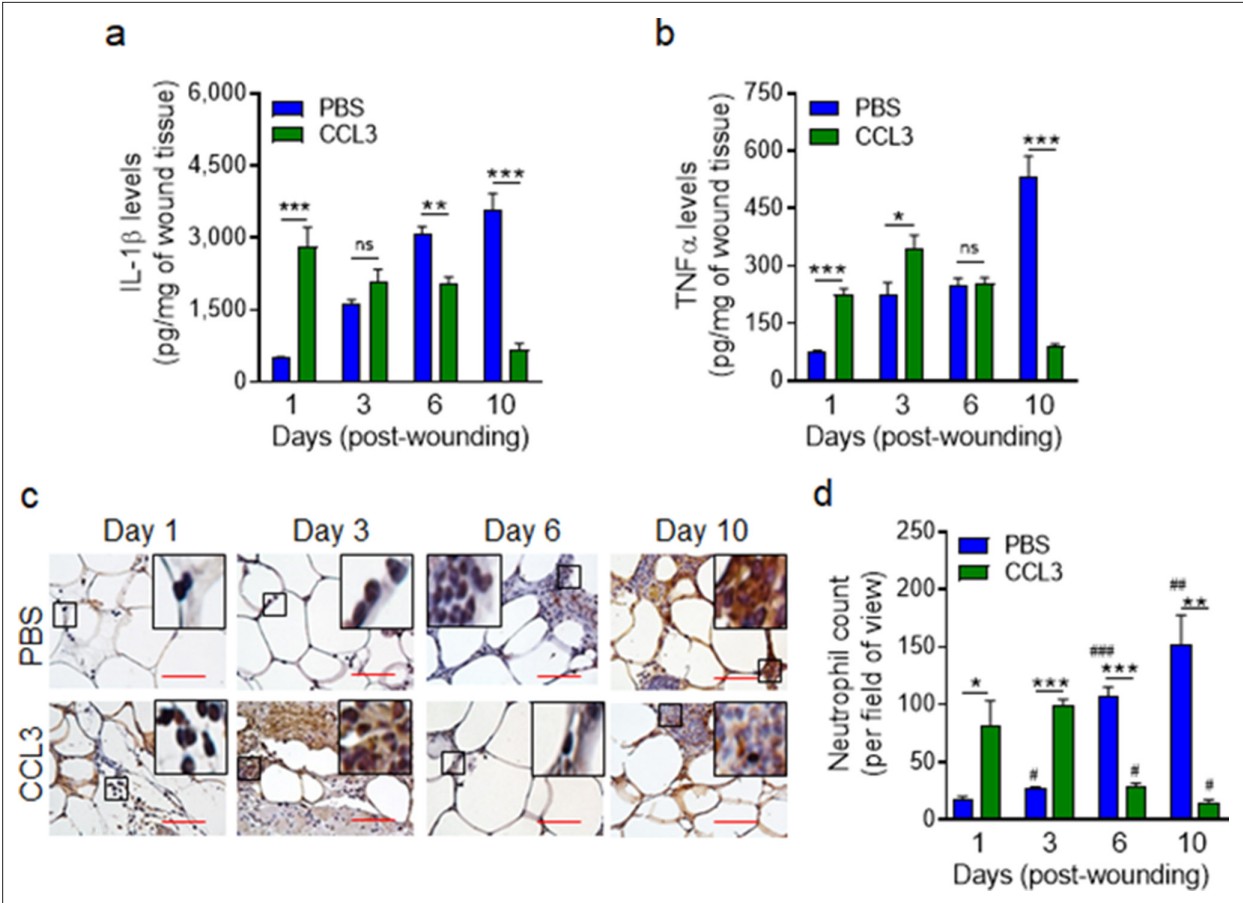

**Figure 5.** Treatment with CCL3 does not lead to persistent inflammation in infected diabetic wounds. db/db wounds were treated with PBS or CCL3 (1 µg/wound) and infected with PA103 (1000 CFU/wound). (**a–b**) Wound tissues were collected at indicated timepoints and assessed for their Il-1β (**a**) and TNF-α (**b**) contents by ELISA. (N = 4 mice/group). (**c–d**) The aforementioned PBS and CCL3-treaded and infected diabetic wounds were assessed for their neutrophil contents by histological analysis using neutrophil marker Ly6G staining. (**c**) Representative images of regions from underneath the wounds extending in the dermis at ×400 magnification are shown. (Red scale bars = 50 µm). Representative full wound images of these staining can be found in *Figure 5—figure supplement 1*. (**d**) The corresponding data were plotted as the Mean ± SEM. (N = 4 mice/group, ≥ 9 random fields/wound/ mouse. (*) denotes significance between groups while (#) indicates significance within the same group in comparison to day 1 of respective wound groups. ns = not significant; *p < 0.05, **p < 0.01, ***p < 0.001, #p < 0.05, ##p < 0.01, ###p < 0.001. Statistical analyses between groups were conducted by One-way ANOVA with additional post hoc testing, and pair-wise comparisons between groups were performed or by unpaired Student's *t*-test).

The online version of this article includes the following source data and figure supplement(s) for figure 5:

**Source data 1.** Related to *Figure 5a*.

**Source data 2.** Related to *Figure 5b*.

**Source data 3.** Related to *Figure 5d*.

**Figure supplement 1.** Full wound images associated with *Figure 5c*.

We assessed the long-term impact of CCL3 treatment on IL-1β and TNF-α pro-inflammatory cytokines that are found to be elevated in chronic diabetic foot ulcers (*Mirza et al., 2013*; *Yan et al., 2016*; *Jeffcoate et al., 2005*). Data indicated that while IL-1β and TNF-α continued to rise in the mock-treated db/db wounds as the diabetic wounds aged, in the CCL3-treated diabetic wounds, these pro-inflammatory cytokines were significantly higher during the acute phase of healing early after injury but declined substantially in old wounds, particularly at day 10 (*Figure 5a–b*). In line with these data, neutrophil wound contents (assessed by histological analysis using neutrophil marker Ly6G staining *Pizza et al., 2005*) were also highly elevated during the acute phase of healing early after injury in the CCL3-treated diabetic wounds but declined significantly as the wounds aged, as compared with the mock-treated diabetic wounds (*Figure 5c–d*, *Figure 5—figure supplement 1*).

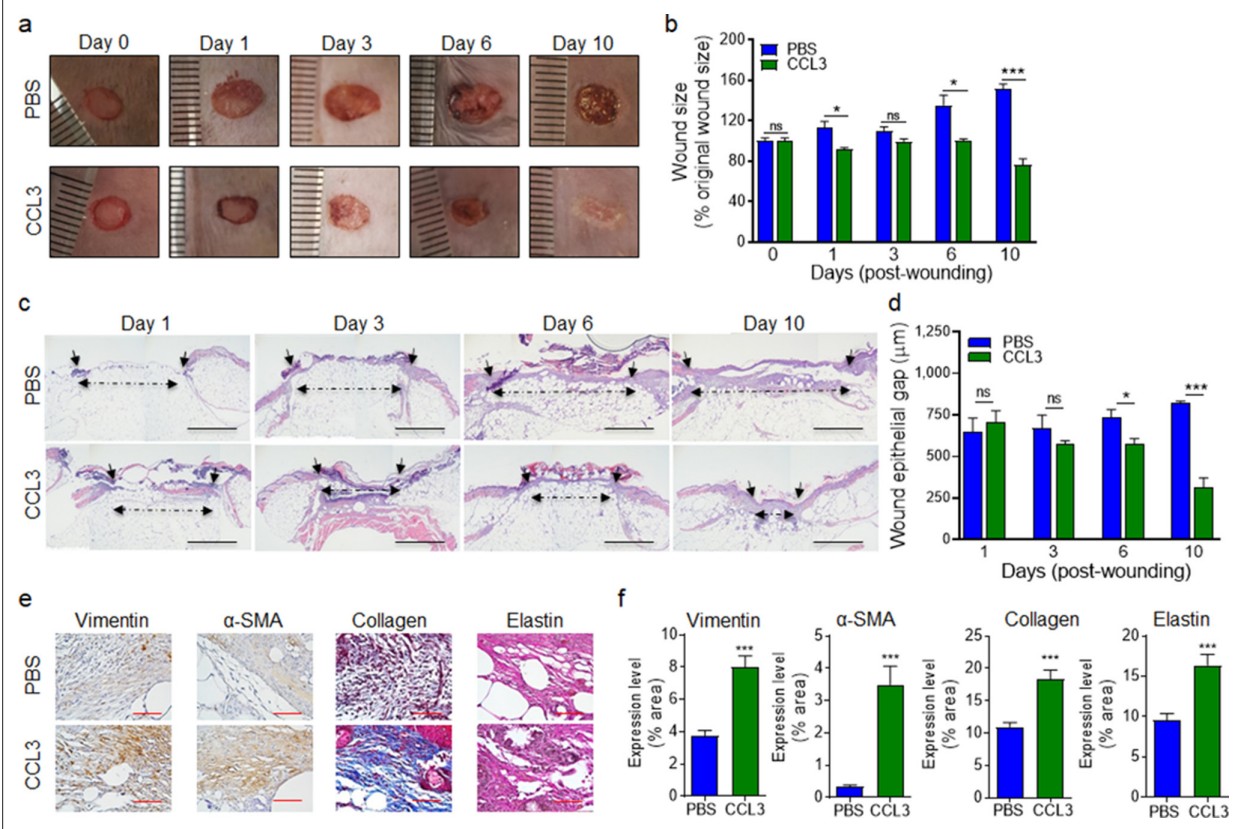

**Figure 6.** Treatment with CCL3 stimulates healing in infected diabetic wounds. (**a–d**) db/db wounds were either treated with PBS or CCL3 and infected with PA103 (1000 CFU). Wound healing was assessed at indicated timepoints by digital photography (**a–b**) or by H&E histological analysis of re-epithelialization (**c–d**). Representative images are shown in (a & c). (Black scale bar = 1 mm, and the wound gap is shown by dotted line). The corresponding data for (a & c) are shown in (b & d) as the Mean ± SEM. (**e–f**) Day 10 db/db wounds (treated with either PBS or CCL3 and infected with PA103) were assessed for fibroblast, myofibroblast, elastin, and cartilage healing markers by vimentin, α-SMA, Masson's Trichrome, and elastin staining, respectively. (**e**) Representative regions from underneath the wounds extending in the dermis are shown at ×400 magnification. (Red scale bar = 50 μm. For the corresponding full wound images at ×40 magnification, see *Figure 6—figure supplement 1*). (**f**) The corresponding data are plotted as the Mean ± SEM. (N = 4 mice/group for (**a–b**); and N = 4 mice/group for (**c–f**). *p < 0.05, **p < 0.01, ***p < 0.001. Statistical analyses between groups were conducted by One-way ANOVA with additional post hoc testing, and pair-wise comparisons between groups were performed or by unpaired Student's *t*-test).

The online version of this article includes the following source data and figure supplement(s) for figure 6:

**Source data 1.** Related to *Figure 6b*.

**Source data 2.** Related to *Figure 6d*.

**Source data 3.** Related to *Figure 6f*.

**Figure supplement 1.** Full wound images associated with *Figure 6e*.

Encouragingly, CCL3 treatment also significantly stimulated healing in infected diabetic wounds, as assessed by wound area measurement (*Figure 6a–b*), while mock-treated diabetic wounds became exacerbated as the result of *P. aeruginosa* infection as we have previously shown (*Goldufsky et al., 2015*). Corroborating these results, CCL3-treated infected diabetic wounds were completely re-epithelized and exhibited epidermal thickening as assessed by H&E histological analysis, while mock-treated infected diabetic wounds became exacerbated (*Figure 6c–d*).

Fibroblasts and myofibroblasts are key players in extracellular matrix production and granulation tissue maturation during the proliferation and the remodeling phases of wound healing (*Wilgus et al., 2008*; *Skalli et al., 1989*; *Cheng et al., 2016*). However, persistent inflammatory environment in diabetic wounds adversely impacts the functions of fibroblast and myofibroblast, culminating in reduced collagen and elastin extracellular matrix deposition and impaired healing in diabetic chronic wounds (*Diegelmann and Evans, 2004*; *Yue et al., 1986*; *Augustine et al., 2014*). *P. aeruginosa*

infection further exacerbates inflammation and reduces collagen deposition in diabetic wounds (*Goldufsky et al., 2015*). We evaluated the impact of CCL3 treatment on fibroblast, myofibroblast, collagen, and elastin in day 10 diabetic wounds, using their respective markers: Vimentin, α-SMA, Elastin, and Masson's Trichrome staining (*Goldufsky et al., 2015*; *Wilgus et al., 2008*; *Hinz, 2006*). CCL3-treated wounds showed significant increases in all these healing markers (*Figure 6e–f* and *Figure 6—figure supplement 1*). Collectively, these data indicate that diabetic wounds are not destined to develop persistent non-resolving inflammation, provided that the dynamics of neutrophil trafficking is restored in these wounds early after injury.

## Discussion

Diabetic wounds are highly susceptible to infection with pathogenic bacteria, such as *P. aeruginosa*, which in turn drive these wounds toward persistent non-resolving inflammation which contributes to impaired healing (*Goldufsky et al., 2015*; *Wetzler et al., 2000*; *Bjarnsholt et al., 2008*). Here, we demonstrate that early after injury, the diabetic wound exhibits a paradoxical and damaging decrease in essential neutrophil trafficking, which in turn renders diabetic wounds vulnerable to infection. Our data point to impaired signaling through FPR (resulting from exposure to high glucose levels), as an important culprit responsible for the delay in the neutrophil response to injury and infection in diabetic wounds.

It is worth noting that 1 hr exposure to high glucose levels dramatically impaired chemotaxis signaling through the FPR in neutrophils, suggesting that even a short-term rise in serum glucose levels could potentially make non-diabetic people transiently immunocompromised and susceptible to infection. In line with this notion, hyperglycemia during the perioperative and postoperative periods are found to be significant risk factors for surgical site infection (SSI) (*Ambiru et al., 2008*; *Sadoskas et al., 2016*), while glycemic control during the perioperative period has been shown to significantly reduce SSI rates both in human and in animals (*Kroin et al., 2015*; *Sadoskas et al., 2016*). It remains unclear why exposure to high glucose dampens the expression and signaling through the FPR. We posit that it may involve metabolic changes, resulting from high glucose in neutrophils. We are actively investigating this possibility.

Our data demonstrate that at least the expression and signaling through CCR1 and CXCR2 auxiliary receptors are not adversely affected by high glucose, but they may not be signaling in diabetic wounds early after injury because of insufficient production of their ligands, such as CCL3. What causes the reduction in the expression of the ligands for these auxiliary receptors in diabetic wounds early after injury remains unclear, but we posit that multiple factors could influence this outcome. One contributing factor could be the reduced influx of neutrophils in diabetic wounds early after injury as we demonstrated here. Leukocytes (including neutrophils) are major cellular sources of these ligands (e.g. CCL3) for auxiliary receptors (*Ridiandries et al., 2018*). Another contributing factor could be the impaired signaling through FPR as our data demonstrated here. FPR activation in neutrophils has been shown to enhance the expression of the ligands in inflammatory and non-inflammatory cells through the production of other pro-inflammatory signaling cues such as IL-1β (*Afonso et al., 2012*; *Chou et al., 2010*; *Su and Richmond, 2015*; *Luster et al., 2005*). Another contributing factor could be reduced TLR signaling in diabetic wounds early after injury due to increased IL-10 expression and signaling (*Roy et al., 2021*). TLR signaling has also been implicated in the production of ligands (e.g. CCL3) for these auxiliary receptors (*Kochumon et al., 2020*; *Ahmad et al., 2019*).

Importantly, one-time topical treatment with CCL3 substantially boosted antimicrobial defenses without leading to heightened non-resolving inflammation in diabetic wounds. These data indicate that diabetic wounds will not develop persistent non-resolving inflammation provided that the neutrophil responses are restored in them early after injury. This finding is consistent with reports highlighting the pivotal role of neutrophils also in the resolution phase of inflammation (through the production and release of anti-inflammatory and inflammation resolving proteins and bioactive lipids, such as Annexin A1, lipoxins, and protectin D1), to ensure that the inflammatory responses cease safely without compromising tissue's defenses against invading pathogens, which they accomplish directly (*Jones et al., 2016*; *Sugimoto et al., 2016*; *Serhan et al., 2008*).

Diabetic chronic wounds are locked in persistent non-resolving inflammation (*Goldufsky et al., 2015*; *Wetzler et al., 2000*; *Bjarnsholt et al., 2008*). Intriguingly, our data indicate that exposure to high glucose causes drastic reduction in both FPR1 and FPR2 expression in neutrophils. Given that

FPR2 has been implicated in the resolution of inflammation in response to Annexin A1, lipoxin A4, and resolving D1 inflammation pro-resolving agonists (*Ye et al., 2009*; *Jeong and Bae, 2020*; *Bena et al., 2012*; *Yazid et al., 2012*; *Serhan and Savill, 2005*), these data suggest that defective signaling through FPR2 in neutrophils may also be a contributing factor to the sustained non-resolving inflammatory environment in chronic diabetic ulcers. Future studies should investigate the role of FPR2 signaling in the resolution of inflammation in acute wound healing and the possibility that defective signaling in the FPR2 may contribute to sustained and non-resolving inflammatory environment in diabetic chronic ulcers.

It remains a possibility that other auxiliary receptors which amplify the neutrophil migration in wounds and toward infection (e.g. CXCR1, BLT1, etc. *de Oliveira et al., 2016*), may also remain functional under diabetic condition and their engagement with their respective ligands could similarly enhance antimicrobial defenses in diabetic wounds. Future studies should address these possibilities and evaluate how serum glucose level affects the expression and/or the activity of all the ~30 receptors that mediate neutrophil chemotaxis in diabetic individuals and toward infection.

It is encouraging that one-time topical treatment with CCL3 after injury also substantially improved healing in diabetic wounds. However, given that diabetic foot ulcers are already suffering from neutrophilia and heightened inflammation, the therapeutic value of CCL3 treatment may seem questionable. We posit that CCL3 topical treatment may have real therapeutic potential in diabetic wound care, at least in a subset of type two obese diabetic individuals represented by our animal model, if applied topically after the surgical wound debridement process. The purpose of surgical debridement, which is performed as a standard-of-care weekly or biweekly in the clinics, is to convert a chronic non-healing wound environment into an acute healing environment through the removal of necrotic and infected tissue, and the senescent and non-responsive cells (*Golinko et al., 2008*; *Lebrun et al., 2010*; *Cardinal et al., 2009*). Therefore, debrided wound environment is likely to be more similar to day 1 fresh wounds than day 10 chronic wounds in our studies. Future studies are needed to evaluate the therapeutic potential of CCL3 in diabetic wound care.

## Materials and methods
### Procedures related to animal studies

We have an approval from the Rush University Medical Center Institutional Animal Care and Use Committee (IACUC No.: 18–037) to conduct research as indicated. All procedures complied strictly with the standards for care and use of animal subjects as stated in the Guide for the Care and Use of Laboratory Animals (Institute of Laboratory Animal Resources, National Academy of Sciences, Bethesda, MD, USA). We obtained 8-week-old C57BL/6 (normal) and their diabetic littermates, C57BLKS-m $Lepr^{db}$ (db/db) mice from the Jackson Laboratories (Bar Harbor, ME). These mice were allowed to acclimate to the environment for 1 week prior to experimentation. Wounding and wound infection were carried out as we described previously (*Goldufsky et al., 2015*; *Wood et al., 2014*). Hematoxylin & Eosin (H&E) staining were performed as we described previously (*Goldufsky et al., 2015*; *Kroin et al., 2016*). Neutrophil trafficking into wounds was assessed by immunohistochemical (IHC) analysis using Ly6G staining as described previously (*Yang et al., 2019*). Wound tissues' contents of myeloperoxidase (MPO) were assessed by ELISA as described (*Kroin et al., 2016*). *CCL3* expression was assessed by RT-PCR, following the protocol we described previously (*Wood et al., 2014*). To account for reduced neutrophil migration into day 1 diabetic wounds, data were normalized by 18 S RNA levels. We used *Pseudomonas aeruginosa* PA103 in these studies. This strain has been described previously (*Shafikhani and Engel, 2006*; *Wood et al., 2015b*) and we have shown that it causes massive infection and exacerbates wound damage in db/db wounds (*Goldufsky et al., 2015*). Infection levels in wounds were evaluated by determining the number of bacteria (colony forming unit (CFU)) per gram of wound tissues, as we described (*Goldufsky et al., 2015*; *Kroin et al., 2015*).

### Histological analyses and wound healing assessment

Wound healing was assessed by digital photography; by re-epithelialization assessment using H&E staining; by fibroblasts and myofibroblasts tissue content analyses using vimentin and α-SMA; and by elastin and collagen matrix deposition assessment using elastin or Masson's Trichrome staining, using previously described techniques (*Goldufsky et al., 2015*; *Wood et al., 2014*; *Roy et al., 2021*;

*Wilgus et al., 2008*; *Almine et al., 2012*; *Diegelmann, 2004*). The histological data were obtained from N = 4 mice/group and ≥9 random fields/wound/mouse. The data were presented as Number of counts per field of view (PFV).

## Neutrophil isolation from human and mouse

We have an Institutional Review Board (IRB)- approved protocol in accordance with the Common Rule (45CFR46, December 13, 2001) and any other governing regulations or subparts. This IRB-approved protocol allows us to collect blood samples from non-diabetic volunteers with their consents for these studies. The blood samples were first checked by a glucometer kit (FreeStyle Lite, Blood Glucose Monitoring System) to ensure that blood glucose level is within the normal range, ≤ 100 mg/dl. Next, human neutrophils were purified from blood using the EasySep Human Neutrophil Enrichment Kit (STEMCELL Technologies), according to manufacturer's protocol.

Murine neutrophils were isolated from either peripheral blood (used in *Figure 2a–c*; *Figure 2— figure supplement 1*; *Figure 3h–i*) or bone marrow (*Figures 2f–m and 3b–g*, *Figure 2—figure supplement 1* and *Figure 2—figure supplements 2 and 3*; and *Figure 3—figure supplement 1*) for the studies involving glucose exposure using EasySep Mouse Neutrophil Enrichment Kit (STEM-CELL Technologies), as per manufacture's protocol and as described previously (*Wood et al., 2014*; *Swamydas et al., 2015*). Mouse neutrophils involving comparisons between C57BL/6 normal and db/db diabetic neutrophils were isolated from N = 4 blood pools/group, with each blood pool being from 4 mice. This was to obtain enough neutrophils from mouse blood (~0.8 ml of blood/mouse, 3.2 ml total) for analyses to achieve statistical significance.

## Neutrophil chemotactic response

Purified human and murine neutrophils were incubated in (IX HBSS with 2% HSA) containing glucose at indicated concentrations for 1 hr at 37 °C and stained with Calcein AM (5 µg/mL) for 30 min. After washing the cells, the cell migration assay was performed in vitro using 96-well disposable chemotaxis chambers (Cat. No. 106–8, Neuro Probe, Gaithersburg, MD, USA). Neutrophils chemotaxis toward the chemoattractants were performed at indicated concentrations, or PBS (to account for the background neutrophil migration), following the manufacturer's protocol. Cell migration was assessed by a fluorescence (excitation at 485 nm, emission at 530 nm) plate reader Cytation 3 Cell Imaging Multi-Mode Reader (Biotek Instruments, Inc). The actual chemotaxis values were obtained by subtracting random chemotaxis values (PBS) from the chemotaxis values in response to indicated ligands.

## Flow cytometry
### Wound tissue digestion and flow cytometric

C57BL/6 and db/db wound tissues were obtained at indicated timepoints as described (*Wood et al., 2014*), weighed, and place immediately in cold HBSS (Mediatech, Inc, Manassas, VA). Subcutaneous fat was removed using a scalpel and scissors were used to cut the tissue into small <2 mm pieces. The tissue was enzymatically dissociated in DNAse I (40 µg/ml; Sigma-Aldrich Co., St. Louis, MO) and Collagenase D (1 mg/ml HBSS; Roch Diagnostics, Indianapolis, IN) at 37 °C for 30 min. Cold PBS was used to stop the dissociation process. The tissue was then mechanically dissociated using the gentleMACS octoDissociator (program B; Miletynyi Biotec, Auburn, CA) and passed through 70 µm nylon screens into 50 ml conical tubes. Cells were washed twice with PBS. Resultant single-cell suspensions were stained using the indicated fluorescently labeled antibodies against cell surface markers, according to standard protocols described previously (*Kohlhapp et al., 2012*; *Zloza et al., 2012*). All antibodies were purchased from eBioscience, Inc (San Diego, CA). Flow cytometry was performed using a the LSRFortessa cell analyzer (Becton, Dickinson, and Company) and data were analyzed using FlowJo software (Tree Star, Ashland, OR), as previously described (*Wood et al., 2014*; *Hackstein et al., 2012*). Briefly, for the gating strategy, Live singlet lymphocytes were identified by gating on forward scatter-area (FSC-A) versus (vs.) side scatter-area (SSC-A), then LIVE/DEAD staining vs. SSC-A, FSC-A vs. FSC-height (H), SSC-A vs. SSC-H, FSC-width (W) vs SSC-W, and CD45 vs SSC-A. T cells, B cells, and NK cells were excluded using antibodies against CD3, CD19, and NK1.1, respectively, all on one channel as a dump gate. Neutrophils were then identified using CD11b vs Ly6G staining, with neutrophils being CD11b high and Ly6G high. FPR1 and CCR1 expression on neutrophils was then analyzed and is presented as percentage of cells (e.g. neutrophils) expressing the respective marker.

## Neutrophil depletion in mice

Neutrophil depletion in mice were performed as described (*Nozawa et al., 2006*; *Bruhn et al., 2016*). Briefly, db/db mice received either anti-Ly6G (100 μg/mouse) to cause neutrophil depletion or an IgG isoform control (100 μg/mouse), by intraperitoneal (i.p.) injection. Neutrophil depletion was confirmed by the assessment of neutrophil content in the blood (circulation) by flow cytometry or in wound tissues by MPO analysis.

## Western blot analyses

We performed western immunoblotting on cell lysates or on tissue lysates, using the indicated antibodies as we described previously (*Kroin et al., 2016*; *Shafikhani and Engel, 2006*; *Mohamed et al., 2021*). Equal amounts of proteins (as determined by BCA analysis) were loaded. GAPDH was used as a loading control.

## Gene expression analysis by real-time polymerase chain reaction (RT-PCR)

Gene expression was assessed by RT-PCR as we described (*Wood et al., 2014*): cDNA was generated using SuperScript III First-Strand Synthesis System cDNA Synthesis Kit (Cat. No. 18080051) from Thermo Fisher, according to manufacturer's protocol. RT-PCR was then preformed with gene-specific primer pairs mentioned below, using the Applied Biosystems QuantStudio 7 Flex Real-Time PCR System. The data were calculated using the $2^{-\Delta\Delta Ct}$ method and were presented as ratio of transcripts for gene of interest normalized to *18*S or *GAPDH*. We performed RT-PCR using the indicated primers listed in the 'Key Resources Table'.

## Statistical analysis

Statistical analyses were performed using GraphPad Prism 6.0 as we described previously (*Roy et al., 2021*; *Wood et al., 2011*; *Wood et al., 2015a*). Comparisons between two groups were performed using Student's *t*-test. Comparisons between more than two groups were performed using one-way analysis of variance (one-way ANOVA). To account for error inflation due to multiple testing, the Bonferroni method was used. Data are presented as Mean ± SEM. Statistical significance threshold was set at -values ≤ 0.05.

# Acknowledgements

We are thankful to Dr. Lena Al-Harthi and Dr. Celeste Napier for the use their equipment. We also would like to thank Mr. Jeffrey Martinson for his help with the flow cytometry and the rest of Shafikhani lab for their valued opinions on these studies. This work was supported by the National Institutes of Health (NIH) grant RO1DK107713 to (SHS), R01AI150668 to (SHS), F31DK118797 to (JZ), and the NIH PhD institutional training grant GM109421.

# Additional information

## Competing interests

Sasha H Shafikhani: Rush University Medical Center has filed a patent (International Application Number: PCT/US19/41112). Dr. Sasha Shafikhani is the listed inventor on this application. The other authors declare that no competing interests exist.

## Funding

| Funder | Grant reference number | Author |
| --- | --- | --- |
| National Institutes of Health | RO1DK107713 | Sasha H Shafikhani |
| National Institutes of Health | R01AI150668 | Sasha H Shafikhani |

| Funder | Grant reference number | Author |
| --- | --- | --- |
| National Institutes of Health | F31DK118797 | Janet Zayas |
| National Institutes of Health | GM109421 | Janet Zayas |

The funders had no role in study design, data collection and interpretation, or the decision to submit the work for publication.

## Author contributions

Ruchi Roy, Data curation, Formal analysis, Investigation, Methodology, Writing – original draft, Writing – review and editing; Janet Zayas, Investigation, Methodology, Writing – original draft, Writing – review and editing; Sunil K Singh, Data curation, Formal analysis, Investigation, Validation; Kaylee Delgado, Formal analysis, Investigation; Stephen J Wood, Mohamed F Mohamed, Dulce M Frausto, Ricardo Estupinian, Eileena F Giurini, Andrew Zloza, Methodology; Yasmeen A Albalawi, Other: Contributed to blood draw and sample preparation; Thea P Price, Other: Contributed to blood draw and sample preparation; Timothy M Kuzel, Jochen Reiser, Conceptualization, Resources; Sasha H Shafikhani, Conceptualization, Data curation, Formal analysis, Funding acquisition, Investigation, Project administration, Resources, Supervision, Writing – original draft, Writing – review and editing

## Author ORCIDs

Ruchi Roy  http://orcid.org/0000-0001-8298-1321
Sasha H Shafikhani  http://orcid.org/0000-0003-1755-9997

## Ethics

Human subjects: We have an Institutional Review Board (IRB)- approved protocol in accordance with the Common Rule (45CFR46, December 13, 2001) and any other governing regulations or subparts. This IRB-approved protocol allows us to collect blood samples from non-diabetic volunteers with their consents for these studies.

We have an approval from the Rush University Medical Center Institutional Animal Care and Use Committee (IACUC No.: 18-037) to conduct research as indicated. All procedures complied strictly with the standards for care and use of animal subjects as stated in the Guide for the Care and Use of Laboratory Animals (Institute of Laboratory Animal Resources, National Academy of Sciences, Bethesda, MD, USA).

## Decision letter and Author response

Decision letter https://doi.org/10.7554/eLife.72071.sa1
Author response https://doi.org/10.7554/eLife.72071.sa2

# Additional files

## Supplementary files
• Transparent reporting form

## Data availability

All data generated or analyzed during this study are included in the manuscript and supporting files. Source data files have been provided for each experiment.

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

# Appendix 1

## Appendix 1—key resources table

| Reagent type (species) or resource | Designation | Source or reference | Identifiers | Additional information |
|---|---|---|---|---|
| Strain, strain background (C57BL/6 J) | C57BL/6 J | Jackson laboratories | 000664 | |
| Strain, strain background (C57BLKS/J) | C57BLKS-m *Lepr*$^{db/db}$ | Jackson laboratories | 000662 | |
| Antibody | Anti-Ly-6G/Ly-6C Monoclonal Antibody (RB6-8C5) (Mouse monoclonal) | Thermo Fisher Scientific | Cat# MA1-10401, RRID:AB_11152791 | For neutrophil depletion (100 µg/mouse) |
| Antibody | Anti-Mouse (G3A1) mAb IgG1 Isotype Control antibody (Mouse monoclonal) | Cell Signaling Technologies | Cat#5415, RRID:AB_10829607 | For neutrophil depletion (100 µg/mouse) |
| Antibody | GAPDH antibody (Rabbit polyclonal) | Proteintech | Cat# 1094-I-AP, RRID:AB_2895245 | WB (1:10000) |
| Antibody | Anti-Ly6G antibody clone RB6-8C5 (Rat monoclonal) | Abcam | Cat# ab25377, RRID:AB_470492 | IHC (1:50) |
| Antibody | Anti-FPR1 antibody (Rabbit polyclonal) | NOVUS Biological | Cat# NB100-56473, RRID:AB_838228 | WB (1:1000) |
| Antibody | Anti-FPR2/ FPRL1 antibody (Rabbit polyclonal) | NOVUS Biologicals | Cat# NLS1878, RRID:AB_2294156 | WB (1:1000) |
| Antibody | Anti-PLC1 antibody (Rabbit polyclonal) | Cell Signaling Technology | Cat# cs2822, RRID:AB_2163702 | WB (1:1000) |
| Antibody | Anti-CCR1 antibody (Rabbit polyclonal) | Abnova | Cat# PAB0176, RRID:AB_1018941 | WB (1:500) |
| Antibody | Anti-α-SMA antibody (Rabbit polyclonal) | Abcam | Cat# ab5694, RRID:AB_2223021 | |
| Antibody | Anti-vimentin antibody (Rabbit monoclonal) | Abcam | Cat# ab92547, RRID:AB_10562134 | |
| Antibody | Mouse CCR1 Alexa Fluor 488-conjugated Antibody (Rat monoclonal) | NOVUS Biologicals | Cat# FAB5986G, RRID:AB_2895246 | Flow cytometry |
| Antibody | Alexa Fluor 700 anti-mouse NK-1.1 Antibody (Mouse monoclonal) | BioLegend | Cat# 108729, RRID:AB_2074426 | Flow cytometry |
| Antibody | Alexa Fluor 700 anti-mouse CD3ε Antibody (Syrian Hamster monoclonal) | BioLegend | Cat# 152315, RRID:AB_2632712 | Flow cytometry |
| Antibody | Alexa Fluor 700 anti-mouse CD19 Antibody (Rat monoclonal) | BioLegend | Cat# 115527, RRID:AB_493734 | Flow cytometry |
| Antibody | BV605 Hamster Anti-Mouse CD11c Clone HL3 (RUO) (Hamster monoclonal) | BD Biosciences | Cat# 563057, RRID:AB_2737978 | Flow cytometry |
| Antibody | F4/80 antibody, Cl:A3-1 (Rat monoclonal) | Bio-Rad | Cat# MCA497PBT, RRID:AB_1102557 | Flow cytometry Flow cytometery |
| Antibody | BV650 Hamster Anti-Mouse CD11c Clone HL3 (Hamster monoclonal) | BD Biosciences | Cat# 564079, RRID:AB_2725779 | Flow cytometry |
| Antibody | BV711 Rat Anti-Mouse CD45 Clone 30-F11 (Rat monoclonal) | BD Biosciences | Cat# 563709, RRID:AB_2687455 | Flow cytometry |
| Antibody | NK1.1 Monoclonal Antibody (PK136), PE, eBioscience (Mouse monoclonal) | Thermo Fisher Scientific | Cat# 12-5941-82, RRID:AB_466050 | Flow cytometry |
| Antibody | CD19 Monoclonal Antibody (eBio1D3 (1D3)), PE, eBioscience (Rat monoclonal) | Thermo Fisher Scientific | Cat# 12-0193-82, RRID:AB_657659 | Flow cytometry |

*Appendix 1 Continued on next page*

*Appendix 1 Continued*

| Reagent type (species) or resource | Designation | Source or reference | Identifiers | Additional information |
|---|---|---|---|---|
| Antibody | CD3e Monoclonal Antibody (145–2 C11), PE, eBioscience (Hamster monoclonal) | Thermo Fisher Scientific | Cat# 12-0031-82, RRID:AB_465496 | Flow cytometery |
| Antibody | FPR1 Polyclonal Antibody (abbit polyclonal) | Thermo Fisher Scientific | Cat# PA1-41398, RRID:AB_2247097 | Flow cytometery |
| Antibody | Goat anti-Rabbit IgG (H + L) Highly Cross-Adsorbed Secondary Antibody, Alexa Fluor 594 (Goat polyclonal) | Thermo Fisher Scientific | Cat# A-11037, RRID:AB_2534095 | Flow cytometery |
| Antibody | Ly6G Monoclonal Antibody (1A8-Ly6g), PE-Cyanine7, eBioscience (Rat monoclonal) | Thermo Fisher Scientific | Cat# 25-9668-82, RRID:AB_2811793 | Flow cytometery |
| Antibody | PerCP Cy5.5 CD45 antibody (Rat monoclonal) | BD Biosciences | Cat# 550994, RRID:AB_394003 | Flow cytometery |
| Antibody | APC Gr1, PE CD11b antibody (Rat monoclonal) | BD Biosciences | Cat# 553129, RRID:AB_398532 | Flow cytometery |
| Antibody | FITC CD69 antibody (Hamster monoclonal) | BD Biosciences | Cat# 557392, RRID:AB_396675 | Flow cytometery |
| Antibody | PECy7 F4/80 antibody (Rat monoclonal) | BioLegend | Cat# 123114, RRID:AB_893478 | Flow cytometery |
| Commercial assay or kit | LIVE/DEAD Fixable Aqua Dead Cell Stain Kit, for 405 nm excitation | Thermo Fisher Scientific | Cat# L34966 | |
| Sequence-based reagent | FPR1_F | Integrated DNA Technologies | RT-PCR primers | GAGCCTAGCCAAGAAGGTAATC |
| Sequence-based reagent | FPR1_R | Integrated DNA Technologies | RT-PCR primers | TCCCTGGTCCAAGTCTACTATT |
| Sequence-based reagent | FPR2_F | Integrated DNA Technologies | RT-PCR primers | TTGTCTCAATCCGATGCTCTATG |
| Sequence-based reagent | FPR2_R | Integrated DNA Technologies | RT-PCR primers | TCAGGGCTCTCTCAAGACTATAA |
| Sequence-based reagent | Plcg1_F | Integrated DNA Technologies | RT-PCR primers | GGTGAGGCCAAATGTGAGATA |
| Sequence-based reagent | Plcg1_R | Integrated DNA Technologies | RT-PCR primers | GGGCAACCAAGAGGAATGA |
| Sequence-based reagent | Ccr1_F | Integrated DNA Technologies | RT-PCR primers | GCTATGCAGGGATCATCAGAAT |
| Sequence-based reagent | Ccr1_R | Integrated DNA Technologies | RT-PCR primers | GGTCCAGAGGAGGAAGAATAGA |
| Sequence-based reagent | Ccl3_F | Integrated DNA Technologies | RT-PCR primers | TCACTGACCTGGAACTGAATG |
| Sequence-based reagent | Ccl3_R | Integrated DNA Technologies | RT-PCR primers | CAGCTTATAGGAGATGGAGCTATG |
| Sequence-based reagent | GAPDH_F | Integrated DNA Technologies | RT-PCR primers | TTGGGTTGTACATCCAAGCA |
| Sequence-based reagent | GAPDH_R | Integrated DNA Technologies | RT-PCR primers | CAAGAAACAGGGGAGCTGAG |
| Sequence-based reagent | 18 S_F | Integrated DNA Technologies | RT-PCR primers | CACGGACAGGATTGACAGATT |
| Sequence-based reagent | 18 S_R | Integrated DNA Technologies | RT-PCR primers | GCCAGAGTCTCGTTCGTTATC |
| Commercial assay or kit | Myeloperoxidase (MPO) Mouse ELISA Kit | Thermo Fisher Scientific | Cat# EMMPO | |
| Commercial assay or kit | IL-1b ELISA kit | Thermo Fisher Scientific | Cat# 88-7013-88 | |

*Appendix 1 Continued on next page*

*Appendix 1 Continued*

| Reagent type (species) or resource | Designation | Source or reference | Identifiers | Additional information |
|---|---|---|---|---|
| Commercial assay or kit | TNF- a ELISA kit | Thermo Fisher Scientific | Cat# 88-7324-88 | |
| Commercial assay or kit | Cyclic AMP Competitive ELISA Kit | Cayman chemical | Cat# 581,001 | |
| Commercial assay or kit | EasySep Human Monocytes Enrichment Kit | STEMCELL Technologies | Cat# 19,359 | |
| Commercial assay or kit | EasySep Mouse monocytes Enrichment Kit | STEMCELL Technologies | Cat# 19,861 | |
| Commercial assay or kit | SuperScript III First-Strand Synthesis System | Thermo Fisher | Cat# 18080051 | |
| Peptide, recombinant protein | CCL3 (recombinant mouse CCL3/MIP-1α protein) | R & D Systems | Cat# 450-MA | |
| Peptide, recombinant protein | N-formyl-Met-Leu-Phe (fMLP) | Sigma | Cat# 59880-97-6 | |
| Peptide, recombinant protein | Recombinant Human CXCL1/ GRO alpha Protein | R & D Systems | Cat# 275-GR | |
| Peptide, recombinant protein | Recombinant Mouse CXCL1/ KC Protein | R & D Systems | Cat# 453-KC | |
| Software, algorithm | GraphPad | GraphPad | https://graphpad.com/ scientific-software/prism/ | |
| Other | Hematoxylin | Thermo Fisher Scientific | Cat# 7,111 L | |
| Other | Eosin Y | Thermo Fisher Scientific | Cat# 7,211 L | |
| Other | Bluing Reagent | Thermo Fisher Scientific | Cat# 7,301 L | |
| Other | Masson's Trichrome stain | Abcam | Cat# ab150686 | |
| Other | EasySep Buffer | STEMCELL Technologies | Cat. No. 20,144 | |
| Other | SYBR Green PCR Master Mix | Thermo Fisher | Cat. No. 4309155 | |
| Other | Collagenase D | Sigma | Cat# 9001-12-1 | |
| Other | Calcein AM | Thermo Fischer Scientific | Cat# C1430 | |

