## [Editor Report]

The data demonstrate substantial neutrophil dysfunction of diabetic or glucose-exposed neutrophils and provide potential therapeutic strategies to improve neutrophil fitness and improve healing of diabetic wounds. The reviewers feel that all their points of concern, suggestions, and comments have been dealt adequately with and that the revised manuscript has improved substantially.

---

## [Decision Letter]

**Decision letter after peer review:**

Thank you for submitting your article "Overriding FPR defective chemotaxis signaling in neutrophils stimulates infection control and healing in diabetic wound" for consideration by *eLife*. Your article has been reviewed by 3 peer reviewers, one of whom is a member of our Board of Reviewing Editors, and the evaluation has been overseen by Carla Rothlin as the Senior Editor. The following individual involved in review of your submission has agreed to reveal their identity: Mauro Perretti (Reviewer #2).

Essential revisions:

They reconfirm earlier findings that glucose renders neutrophils less responsive to fMLF-mediated chemotaxis and show that expression and surface presentation of the corresponding receptor FPR1, a receptor that is high in the signaling hierarchy, is downregulated within the first hour of glucose treatment. Similarly, other elements of neutrophil chemotactic responses including the phospholipase PLC and the cytokine MIP-1α/CCL3 are also affected, while the expression of the chemokine receptor CCR1 remains unaltered. Interestingly, supplementing the CCFR1-targeting cytokine CCL3 could restore neutrophil chemotactic fitness and wound healing and thus, might be beneficial for diabetic wound management.

All three reviewers agree that the research area is important, the findings are novel and interesting and of potential therapeutical value. While the study is well planned and the results are logically presented, two of the reviewers feel that key aspects of the study require additional data to fully support the central conclusions and that the results at the current stage provide an only limited increase in knowledge on metabolic regulation of neutrophil functions. We, therefore, encourage the authors to perform the requested experiments to strengthen their claims.

Experimental work:

1) A major point of concern is that the very low and sometimes unclear n values question the validity and robustness of the data. Please provide power analyses to justify the group sizes. Furthermore, the n number needs to be stated fully in every figure legend as well as in the Methods section, in line with *eLife* policies. The statement that there are 16 mice per neutrophil comparison group is confusing, as the blood of 4 mice has been pooled. Therefore, one pool is more likely n=1, please address this issue. Also, please correct n values in figure legends according to the information given in the Methods ≥ 5 mice/group.

2) The fact that FPR1 mRNA and FPR1 protein rapidly decrease within 1 hour of high glucose treatment is astonishing and no mechanistic explanation is provided. As there are two homologous FPR receptors responding to formylated peptides (although to different levels of sensitivity/specificity) in human as well as in murine neutrophils, the authors should analyze the expression pattern of both FPR1 and FPR2 in diabetic and glucose-exposed human and murine neutrophils. Because no proof is provided that *P. aeruginosa* bacterial products (or any other FPR1 ligands) acting via FPR1 are responsible for the observed neutrophil recruitment in healthy mice or the lack of in diabetic mice, it should be addressed whether the downregulation is a more common phenomenon and not restricted to FPR1 This question should be addressed, especially as the title emphasizes the importance of FPR1. Is the diminished cell surface pool due to receptor internalization? Please extend the time course and include days 1, 3, 6 and 10 for a more in-depth analysis of neutrophil changes (FPR, CCL3) and effects of topical treatment to enhance the neutrophil response (figure 4) in in diabetic wounds over time. Are the findings specific for neutrophils or do e.g. monocytes respond similarly?

Methodology:

1) Sex differences have been shown to be important for the diabetic wound healing. Please provide whether male or female mice have been used and the rational for choice of sex and age.

2) As both bone marrow-derived neutrophils and blood neutrophils are used, please provide the reasons for the respective choice of the very different compartments and verify that neutrophils derived from these two compartments behave in a similar manner regarding receptor expression and downregulation. Please include the baseline cells counts (especially neutrophils) for the mice used in the study.

3) Please provide and discuss the exact RFU numbers in Figures S3 and Figure 2f.

Discussion:

Please elaborate on the relevance of your specific findings with regard to (1) the mechanistic basis (CCL3 source? Molecular regulation underlying downregulation on the different expression levels?) and their relevance in the more general picture of the link between onset and resolution of inflammation as discussed in e.g. [PMID: 16369558 DOI: 10.1038/ni1276, PMID: 27199985 PMCID: PMC4845539 DOI: 10.3389/fimmu.2016.00160, PMID: 27021499 DOI: 10.1016/j.smim.2016.03.007].

*Reviewer #1 (Recommendations for the authors):*

In human as well as in murine neutrophils, there are two receptors (FPR1 and FPR2) that react to formylated peptides, although to different levels of sensitivity/specificity. While the human FPR1 is highly sensitive to fMLF, FPR2 is more reactive towards host peptides, and both receptors also detect mitochondrially derived peptides. In mice, the sensitivity toward fMLF is generally low and other formylated peptides exert stronger chemotactic activities via FPRs. Commonly, FPR1 is thought to function in a pro-inflammatory manner, whereas FPR2 activation is linked to the resolution phase. The authors should analyze the expression pattern of both FPR1 and FPR2 in diabetic and glucose-exposed human and murine neutrophils and also include other ligands. Furthermore, the authors assume that *Pseudomonas* aeruginosa-infected wounds in diabetic mice show slower healing because the neutrophils are FPR1-deficient and therefore less attracted toward the infection site. However, no proof is provided that *P. aeruginosa* bacterial products (or any other FPR1 ligands) acting via FPR1 are responsible for the observed neutrophil recruitment in healthy mice or the lack of in diabetic mice. This question should be addressed, especially as the title emphasizes the importance of FPR1. It is also conceivable that another (unknown) receptor is downregulated under diabetic conditions.

*Reviewer #2 (Recommendations for the authors):*

Roy et al., present an interesting manuscript where the molecular and cellular responses are studied in a model of diabetes wound healing, or lack of, and infection. The study is well planned and logically presented.

The most interesting aspect of the presented work, to me, is the clear link between an effective and exuberant inflammatory reaction and resolution with tissue repair. The ability of CCL3 application to restore a proper neutrophil response to the infected wounds and as such ensure the appropriate management of the infection is an excellent proof that 'the beginning programs the end' [PMID: 16369558 DOI: 10.1038/ni1276]. On a similar vein, dissecting how resolution starts is equally challenging and interesting [PMID: 27199985 PMCID: PMC4845539 DOI: 10.3389/fimmu.2016.00160] and well presented here. Finally, the notion that neutrophils are not just BAD is also excellent and the study here provides a very clear evidence for the link between onset and resolution [PMID: 27021499 DOI: 10.1016/j.smim.2016.03.007]. This issue may need to be highlighted in the Discussion section as chiefly demonstrated here.

I have some comments and some unclear issues I would like the Authors to comment or address.

1. FPR1 mRNA (Figure 2)

I was surprised to see the rapid decade in FPR1 mRNA within 1 hour of high glucose. To my knowledge, FPR1 is not an early response gene (like fos and jun) so it will be appropriate for the Authors to perhaps double check this dataset, perhaps running a time course and study what else goes down within 30-60-90-120 min. Would there be mileage in monitoring FPR2 mRNA too as a relevant control?

2. FPR1 protein

The same question can be asked for the rapid downregulation of FPR1 protein from the cell surface of mouse and human neutrophils following incubation with glucose. The data are convincing here, but again very intriguing how this could occur. More speculation in the Discussion would be good. Also, experiments aiming to assess if the receptor internalised (thus do we find higher intracellular levels?) and if so, how, would be appropriate for a study published in *eLife*.

3. Legends to the Figures

I find the figure legends written in an unclear manner. usually the Panel letter (e.g. A, or B or C…) is at the beginning of the sentence, whereas here is all a bit mixed up. In some cases groups of 3 mice have been used. Is this acceptable? How reproducible are the Western blots?

4. CCL3

Data in Figure 4C seem to indicate that the infiltrating neutrophils are responsible for delivering CCL3 (or MIP-1alpha). Probably some indication of the potential sources of CCL3 in the infected wounds, including the migrated neutrophils, is worthwhile. And relevant and interesting.

*Reviewer #3 (Recommendations for the authors):*

The manuscript by Roy et al. entitled "Overriding defective FPR chemotaxis signaling in diabetic neutrophil stimulates infection control in diabetic wound", focuses on impaired healing in diabetic wounds. Moreover, the role that neutrophils play and the therapeutic potential of the pro-inflammatory cytokine CCL3 in the treatment of diabetic foot ulcers. Overall, this is an interesting manuscript, but there are several points which should be addressed:

• The n values for this study seem low, n=2 or 2 in some cases. This reviewer would argue the validity and robustness of performing statistical analysis on such low n numbers and the ultimate value of the findings.

• In every figure legend and methods section, the authors need to state fully the n number to provide transparency in their data and in line with *eLife* policies.

• Methods section: Neutrophil isolation from human and mouse. "Mouse neutrophils involving comparisons between C57B normal and db/db diabetic neutrophils were extracted from N=4 blood pools/group, with each blood pool being from 4 mice: totaling 16 mice per group." This is incorrect. There are not 16 mice per group. If each group has blood pooled from 4 mice, then this is n=1, not 4, therefore n=4, not 16. This needs to be addressed.

• Methods section: Histological analyses and wound healing assessment. "The histological data, (obtained from n≥5 mice/group and >9 random fields/wound/mouse), were normalized per wound surface area". Although the authors state ≥5 mice/group, the figure legends suggest differently. This needs to be corrected.

• What are the baseline cells counts (especially neutrophils) for the mice used in the study. These should be included.

• In some cases, the authors use bone marrow derived neutrophils and in other cases blood borne neutrophils, although it is unclear to this reviewer what determined the choice. As it is known that these compartments are very different, it would add value to the manuscript to show whether neutrophils derived from these two compartments behave differently and express different cell surface receptors. Furthermore, these changes/similarities should be shown over the time course of 1, 3, 6 and 10 days, as this may help to further explain what is causing the delay in neutrophil influx in diabetic wounds over time.

• Although CCL3 expression is reduced in day 1 diabetic wounds, what happens at 3, 6 and 10 days?

• The finding that topical treatment enhances neutrophil response (figure 4) is very interesting. These data are for day 1, what happens over the time course i.e. 3, 6 and 10 days?

• The authors show nicely the involvement of FPR1, but what about the other known FPR that neutrophils express i.e. FPR2? Does this have a compensatory role or is this receptor also dysfunctional. Data should be provided.

• In addition, what happens to FPR1 (and 2) expression over the time course i.e. 3, 6 and 10 days? Does it change?

• Were the mice male or female or a mixture? Please provide rational for choice and whether sex as a biological variable was considered, especially given the fact that gender differences have been shown in recovery to diabetic wounds.

• The authors chose mice at 8 weeks, with 1 week acclimatisation. Given this young adult age, what is the clinical relevance of this choice of age?

• Figure S3: 100nM fMLP: The data for the 90 mg/dl is around 800 RFU, however, in Figure 2f, this value looks to be much higher (around 1200 RFU). Would the authors provide the exact numbers and discuss further.

• Although this student specifically looks at neutrophils, do the authors have any data to suggest whether these finds are neutrophil specific or whether e.g. monocytes, play a role in the later time points?

• C57/B should be written in full: C57Bl/6. 'FPR receptor' should be changed to just 'FPR'. Proof reading would help – grammar and spelling mistakes need correcting.

---

## [Author Response]

Essential revisions:They reconfirm earlier findings that glucose renders neutrophils less responsive to fMLF-mediated chemotaxis and show that expression and surface presentation of the corresponding receptor FPR1, a receptor that is high in the signaling hierarchy, is downregulated within the first hour of glucose treatment. Similarly, other elements of neutrophil chemotactic responses including the phospholipase PLC and the cytokine MIP-1α/CCL3 are also affected, while the expression of the chemokine receptor CCR1 remains unaltered. Interestingly, supplementing the CCFR1-targeting cytokine CCL3 could restore neutrophil chemotactic fitness and wound healing and thus, might be beneficial for diabetic wound management.All three reviewers agree that the research area is important, the findings are novel and interesting and of potential therapeutical value. While the study is well planned and the results are logically presented, two of the reviewers feel that key aspects of the study require additional data to fully support the central conclusions and that the results at the current stage provide an only limited increase in knowledge on metabolic regulation of neutrophil functions. We, therefore, encourage the authors to perform the requested experiments to strengthen their claims.Experimental work:1) A major point of concern is that the very low and sometimes unclear n values question the validity and robustness of the data. Please provide power analyses to justify the group sizes. Furthermore, the n number needs to be stated fully in every figure legend as well as in the Methods section, in line with eLife policies. The statement that there are 16 mice per neutrophil comparison group is confusing, as the blood of 4 mice has been pooled. Therefore, one pool is more likely n=1, please address this issue. Also, please correct n values in figure legends according to the information given in the Methods section ≥ 5 mice/group.

First, we would like to thank the reviewers and the editor for their thorough and insightful reviews of our manuscript. Addressing their suggestions have substantially improved our manuscript as we hope the editor and the reviewers will agree. (New data added to revised manuscripts are Figure 5a-b, Figure S4a-c, Figure 2—figure supplement 1 and Figure 2—figure supplement 1). As for responding to query 1, we sincerely apologize to the reviewers and the editor for this confusion in the figure legends. Nowhere in our manuscript did we rely on N=2 for our analyses and we completely agree with the reviewers that N=2 is not statistically robust. We indicated N=2 for experiments involving RT-PCR but we had repeated these experiment at least two independent times so the actual numbers are N>4. We also apologize to the reviewers for indicating N=16 in the animal experiments where blood from 4 mice were pooled together to be able to obtain enough neutrophils. We agree with the reviewer that in reality N=4 blood pools per group (generated from 16 mice/group). As requested, we have made the corrections of statistical analyses both in the figure legends and in the Methods section of the revised manuscript. In addition, we have provided the raw data for all the figures so that anyone who wishes to examine the data will be able to do so.

2) The fact that FPR1 mRNA and FPR1 protein rapidly decrease within 1 hour of high glucose treatment is astonishing and no mechanistic explanation is provided. As there are two homologous FPR receptors responding to formylated peptides (although to different levels of sensitivity/specificity) in human as well as in murine neutrophils, the authors should analyze the expression pattern of both FPR1 and FPR2 in diabetic and glucose-exposed human and murine neutrophils. Because no proof is provided that *P. aeruginosa* bacterial products (or any other FPR1 ligands) acting via FPR1 are responsible for the observed neutrophil recruitment in healthy mice or the lack of in diabetic mice, it should be addressed whether the downregulation is a more common phenomenon and not restricted to FPR1 This question should be addressed, especially as the title emphasizes the importance of FPR1. Is the diminished cell surface pool due to receptor internalization? Please extend the time course and include days 1, 3, 6 and 10 for a more in-depth analysis of neutrophil changes (FPR, CCL3) and effects of topical treatment to enhance the neutrophil response (figure 4) in in diabetic wounds over time. Are the findings specific for neutrophils or do e.g. monocytes respond similarly?

We thank the reviewers and the editor for their recognition of the importance of our data. As the reviewers and the editor astutely pointed out, (both in human and mouse), FPR1 has a much higher affinity for bacterial formyl peptides than FPR2, whereas, FPR2 has a broader range of ligands than FPR1 and it can also respond to other ligands, such as amyloid peptides, antimicrobial peptides, and lipid mediators (1, 2). In addition, FPR2 receptor has also been implicated in the resolution of inflammation in response to Annexin A1, lipoxin A4, and resolving D1 pro-resolving agonists (3-5). Given that this manuscript was about diabetic neutrophil’s impaired chemotactic response toward infection, we had focused on FPR1 receptor in this manuscript. With that said, we now provide new data to show that exposure to HG also significantly dampens the expression of FPR2 receptor in neutrophils both at the transcriptional and translational levels (please see Figure 2—figure supplement 3). We also provide new data to show that exposure to HG substantially reduces both FPR1 and FPR2 at 1, 2, and 3 hours post exposure (please see Figure 2—figure supplement 2 and Figure 2—figure supplement 3), indicating that the adverse impact of HG on FPR1 expression is not transient and it is sustained. We are in complete agreement with the reviewers and the editor that our manuscript does not provide direct evidence that the recognition of *P. aeruginosa* by neutrophils is primarily mediated by the FPR1 and because of this reason, we have revised the manuscript to take into account the reviewers and the editor’s suggestion, by de-emphasizing FPR1. Rather we use the term FPR instead.

As for the suggestion to extend the time course of neutrophil characterizations in normal and diabetic wounds to days 3, 6, and 10, in addition to day 1 data that we had provided, we believe that expanding neutrophil characterizations beyond day 1 would go beyond the scope of this manuscript and could also potentially raise other interesting questions that may detract from the main point of this manuscript which is the novel finding that inadequate neutrophil response early after injury, renders diabetic wounds vulnerable to infection. We are very much interested in addressing the questions raised by the editor and the reviewers such as whether the impact of HG on FPR signaling is limited to neutrophils or all leukocytes. Although, we do believe that with respect to infection control, our data clearly show that the neutrophils play the primary role in controlling *P. aeruginosa* infection in wound (please see Figure 4), therefore, we focused on neutrophils. We also agree that it would also be interesting to assess the dynamics of neutrophil surface expression of chemotaxis receptors and their ligands (such as CCL3) during early and late phases of healing in normal and diabetic wounds. However, we believe that such studies would require a more comprehensive analyses on all ~30 chemotaxis receptors on neutrophils and their respective ligands. In addition, we would also need to characterize the wounds’ environmental factors that can potentially enhance or dampen the expression of these receptors and/or their signaling pathways in neutrophils in wounds at these timepoints. We would also need to validate our results. As the editor and the reviewers appreciate, these studies are beyond the scope of any single manuscript. In fact, we have submitted a grant proposal to conduct these comprehensive studies and also to tease out the mechanism (s) that is responsible for the HG-induced reduction in the FPR receptor signaling in diabetic neutrophils. We acknowledge that our manuscript does not address all the interesting questions that were raised by the reviewers and the editor, but our data do reveal important information regarding the culprit responsible for the impaired chemotaxis responses in diabetic neutrophils, which had been known but ignored for decades (10), and how this impairment affects the dynamics of diabetic neutrophil trafficking into wound during the acute phase of healing, early after injury. Our data clearly establish a new paradigm that blames inadequate neutrophil response early after injury for rendering diabetic wounds vulnerable to infection and for setting the stage for the sustained and non-resolving inflammatory environment during the chronic phase as diabetic wounds age (11, 12). We also show that neutrophil depletion in diabetic animals by anti-Ly6G causes diabetic wounds to contain significantly more bacteria, indicating that as impaired as diabetic neutrophils may be with respect to their bactericidal functions as has been reported (11, 12), they still maintain some degree of antimicrobial functions under diabetic conditions. Finally, we show that by harnessing these neutrophils in diabetic wounds early after injury by topical treatment with CCL3, which engages CCR1, CCR4, and CCR5 auxiliary receptors (13-15), we can reduce infection levels in diabetic wounds by ~2 log orders and significantly improve healing in diabetic wounds.

As the editor and the reviewers are aware, there is only one FDA-approved therapy (Becaplermin) showing modest effectiveness in stimulating wound healing in diabetic wounds (16-21), and there are no treatments to address infection in diabetic wounds, other than the use of systemic antibiotics which are routinely included in the management of diabetic patients with chronic ulcers (22, 23). Antibiotic overuse can have disastrous consequences, leading to the emergence and the spread of antibiotic resistance, cytotoxicity, allergic reactions, and immunological and neurological diseases (24-29). Therefore, our data are also very important as they reveal therapeutic potential for CCL3 topical treatment to enhance infection control and stimulate healing in diabetic wounds. We hope that the reviewers and the editor agree with us that no single manuscript can possibly address all questions, but impactful manuscripts, as we believe our manuscript is, often also open new areas for follow-up research.

Methodology:1) Sex differences have been shown to be important for the diabetic wound healing. Please provide whether male or female mice have been used and the rational for choice of sex and age.

We appreciate and agree with the reviewer #3 that sex is a critical biological variable that needs to be considered in designing any experiment. In studies involving human neutrophils, we isolated neutrophils from both male and female volunteers, and regardless of their sex, these neutrophils behaved similarly with respect to their chemotactic responses toward fMLF and CCL3. In addition, neutrophils isolated from male and female volunteers also exhibited the same reduction in FPR signaling when exposed to HG, suggesting that sex does not affect neutrophil chemotaxis signaling, at least in cell culture. However, since several prior publications have indicated that type 2 diabetic mellitus is more common in males than females [100-103], and because of the budgetary constraints and the high expenses associated with type 2 db/db diabetic mice, we restricted our analysis to male db/db mice.

2) As both bone marrow-derived neutrophils and blood neutrophils are used, please provide the reasons for the respective choice of the very different compartments and verify that neutrophils derived from these two compartments behave in a similar manner regarding receptor expression and downregulation. Please include the baseline cells counts (especially neutrophils) for the mice used in the study.

We appreciate and agree with the reviewers that bone marrow neutrophils and blood neutrophils are not the same but our decision to use neutrophils from these different sources was deliberate because we intended to address different questions using these neutrophils. The initial observation – that showed diabetic neutrophils had impaired chemotaxis – involved neutrophils that were isolated from the blood of diabetic patients (10). Therefore, we isolated neutrophils from the blood of diabetic mice to show that the same chemotaxis impairment was also present in neutrophils isolated from the blood of diabetic mice, as a way to reproduce the human diabetic condition with respect to this impairment. On the other hand, when our goal was to assess the impact of high glucose on chemotaxis and chemotaxis signaling in neutrophils, we isolated neutrophils from the bone marrow of normal mice and exposed them to HG to show that exposure to HG was sufficient to cause similar chemotaxis impairment in normal neutrophils. Of note, bone marrow leukocytes (including neutrophils) in diabetic animals do not appear to have the same functional impairments as peripheral blood leukocytes in diabetic animals or human, as they behave similarly to leukocytes extracted from bone marrow of normal mice (11, 32-34), suggesting that bone marrow leukocytes in diabetic animals are protected from the adverse effects of exposure to HG in the bone marrow compartment. As the reviewers and the editor are aware, neutrophils are produced in bone marrow and released into circulation (30, 31), hence our rationale for using bone marrow as the source of normal neutrophils. Finally, it was also more practical to use bone marrow neutrophils, because we could obtain significantly more neutrophils from bone marrow than blood of these mice. As we had indicated in our manuscript, we had to pool the blood of 4 mice (16 mice/group) to be able to obtain enough neutrophils for a single analysis. As for the request to include the baseline neutrophil counts in our animal studies, we believe that these are not necessary for the following reasons. Neutrophils are rarely found in the skin and are not considered skin-resident cells, but they migrate into the skin in high numbers in response to inflammatory conditions, such as injury and/or infection (35-39). Consistent with these reports, we have also demonstrated that in the absence of infection and injury, neutrophils are not detected in the skin of mice (37). Given that we did not keep the skins of the mice at T0 (time of wounding), we would need to repeat these studies to generate the requested data. We hope that the reviewers and the editor agree with us that including the baseline numbers on neutrophils would not add any new information or change any of the figures or their interpretations and thus repeating these studies is not warranted, particularly because both co-first authors on this manuscript have left the lab.

3) Please provide and discuss the exact RFU numbers in Figures S3 and Figure 2f.

As the reviewer appreciates, many factors can cause intrinsic variations in data, including the operator and/or source of reagents. In our experience, even using the fluorescent dye from the same vial and at same concentration could lead to different intensities and RFU numbers in replicate experiments, particularly if the experiments were performed by different individuals, which was the case for Figure 2f and Figure 2—figure supplement 1. The important point is that in both experiments (which they had their own appropriate controls), we did obtain similar significant results showing that neutrophils treated with normal glucose level (90 mg/dl) displayed significantly higher chemotaxis as compared to neutrophils treated with HG. As requested by the reviewer, we provide the raw data and the exact RFUs for Figure 2f (100nM fMLF) and Figure 2—figure supplement 1 (at indicated fMLF concentrations). We have included a file called “Source data” that contain the raw data used in all the figures should the reviewers wish to evaluate them.

Discussion:Please elaborate on the relevance of your specific findings with regard to (1) the mechanistic basis (CCL3 source? Molecular regulation underlying downregulation on the different expression levels?) and their relevance in the more general picture of the link between onset and resolution of inflammation as discussed in e.g. [PMID: 16369558 DOI: 10.1038/ni1276, PMID: 27199985 PMCID: PMC4845539 DOI: 10.3389/fimmu.2016.00160, PMID: 27021499 DOI: 10.1016/j.smim.2016.03.007].

We thank the reviewer for his/her knowledge of the field and for bringing to our attention these articles, particularly Jones, et al. article (2016, Seminars in Immunology), which discusses the role of neutrophils in inflammation resolution. We have revised the Discussion section to address the reviewer’s request. Regarding the mechanistic basis for CCL3 cellular sources and the reduction in CCL3 expression in diabetic wounds early after in jury, we posit that multiple factors could contribute to the reduction in CCL3 in diabetic wound early after injury. As we discussed in the introduction section, production of ligands (including CCL3) for auxiliary receptors in tissue ultimately depends on FPR activation in neutrophils, resulting in the production of IL-1b which induces the production of these auxiliary ligands from inflammatory and non-inflammatory cells (40-43). Therefore, reduced signaling through FPR, would be expected to adversely affect the expression of the ligands (including CCL3) for auxiliary receptors in diabetic wound early after injury. In addition, leukocytes (including neutrophils) are major cellular sources of CCL3 (44). Therefore, we speculate that reduction in CCL3 expression in diabetic wound early after injury is at least in part due to reduced neutrophil influx in these wounds, as we show in this manuscript (Figure 1). Moreover, we just published a paper in the Journal of Investigative Dermatology (45), showing that dysregulation in the expression of immunosuppressive IL-10 (high IL-10 expression in diabetic wounds early after injury but low IL-10 expression in old diabetic wounds) leads to significant reduction in toll-like receptor (TLR) signaling and culminates in significant reduction in the production of pro-inflammatory cytokines in diabetic wounds early after injury. Given that TLR signaling has also been implicated in the production of ligands (including CCL3) for auxiliary receptors expression (46, 47), we posit that reduced TLR signaling could also contribute to reduced CCL3 expression in diabetic wound early after injury. As the editor and the reviewers appreciate, substantial amount of work is needed to teas out the contributions of these factors in the dynamics of CCL3 and other ligands for the auxiliary receptors, which is beyond the scope of this manuscript. Nevertheless, we have added a section in the Discussion section of the revised manuscript to discuss these possibilities.

As for the link between our data (Figure 4) and the onset inflammatory responses in the CCL3-treated wounds, we expected this outcome, given that CCL3 is an important pro-inflammatory cytokine which has been shown to recruit neutrophils by engaging multiple auxiliary receptors, such as CCR1, CCR4, and CCR5 (40, 43, 48-50). As for the resolution of inflammatory responses in the CCL3-treated diabetic wounds during the late stages of healing in diabetic wounds, it could be due of reduction in infection and bacterial burden in CCL3-treated diabetic wounds as we have shown here (Figure 4h-j), thus reducing the need for neutrophils to combat infection; but it also could be due to the production and release of anti-inflammatory and inflammation resolving proteins and bioactive lipids by recruited neutrophils, such as Annexin A1, lipoxins (e.g., LXA4), and protectin D1 (reviewed in (51)). In addition, uptake of neutrophil apoptotic bodies by macrophages through efferocytosis has also been shown to induce the M2 anti-inflammatory macrophage differentiation which further resolve inflammation. We have added a section in the Discussion section of the revised manuscript regarding this.

Reviewer #1 (Recommendations for the authors):In human as well as in murine neutrophils, there are two receptors (FPR1 and FPR2) that react to formylated peptides, although to different levels of sensitivity/specificity. While the human FPR1 is highly sensitive to fMLF, FPR2 is more reactive towards host peptides, and both receptors also detect mitochondrially derived peptides. In mice, the sensitivity toward fMLF is generally low and other formylated peptides exert stronger chemotactic activities via FPRs. Commonly, FPR1 is thought to function in a pro-inflammatory manner, whereas FPR2 activation is linked to the resolution phase. The authors should analyze the expression pattern of both FPR1 and FPR2 in diabetic and glucose-exposed human and murine neutrophils and also include other ligands. Furthermore, the authors assume that *Pseudomonas aeruginosa*-infected wounds in diabetic mice show slower healing because the neutrophils are FPR1-deficient and therefore less attracted toward the infection site. However, no proof is provided that *P. aeruginosa* bacterial products (or any other FPR1 ligands) acting via FPR1 are responsible for the observed neutrophil recruitment in healthy mice or the lack of in diabetic mice. This question should be addressed, especially as the title emphasizes the importance of FPR1. It is also conceivable that another (unknown) receptor is downregulated under diabetic conditions.

We appreciate reviewer’s knowledge of the field and precisely because of the differences between FPR1 and FPR2, as noted by the reviewer, we had focused our analysis on FPR1, given that this manuscript involves the role of impaired chemotaxis in diabetic neutrophils in response to *Pseudomonas aeruginosa* infection. Per reviewer’s request, we now provide new data which show that HG significantly dampens the expression of both FPR1 and FPR2 (please see Figure 2—figure supplement 2 and Figure 2—figure supplement 3). These data are interesting as they suggest that reduced FPR1 signaling may be an important factor in reduced neutrophil trafficking into diabetic wounds early after injury. In contrast, reduced signaling through FPR2 may be an important contributing factor in the persistent non-resolving inflammatory environment in old and chronic diabetic wounds, which contain a lot of neutrophis, as have been reported (52-54) and confirmed by our data (Figure 1). We acknowledge that we do not provide direct evidence that *P. aeruginosa* is primarily recognized by FPR1, nor do we believe that FPR1 is the only receptor responsible for *P. aeruginosa* recognition by innate immune responses in wounds. However, as the reviewer is aware, FPR signaling plays a crucial role in the initial neutrophil wave toward injury and infection (40, 42, 55-59), and it is therefore logical to posit that defective signaling through FPR in diabetic wound plays an important role in reduced neutrophil influx in diabetic wound early after injury as we show in this manuscript. We also agree with the reviewer that it would be important to assess the importance of FPR1 in the recognition and control of *P. aeruginosa* infection in normal and diabetic mice but we believe that such comprehensive studies would require substantial amount of additional work, which as the reviewer appreciates would go beyond the scope of this manuscript.

Reviewer indicated “In mice, the sensitivity toward fMLF is generally low and other formylated peptides exert stronger chemotactic activities via FPRs”. We apologize to the reviewer if we have misinterpreted the reviewer’s comments which suggest to us that bacterial fMLF recognition by FPR may not play a meaningful physiological role during infection. However, if our interpretation of the reviewer’s comment is correct, we respectfully disagree with the reviewer. FPR knockout mice are highly vulnerable to bacterial infection, even in animal models that do not involve wounds or surgery (60, 61), suggesting that bacterial recognition (most likely bacterial formyl peptides) by FPR is likely to play an important role in combating infection.

Reviewer #2 (Recommendations for the authors):Roy et al., present an interesting manuscript where the molecular and cellular responses are studied in a model of diabetes wound healing, or lack of, and infection. The study is well planned and logically presented.The most interesting aspect of the presented work, to me, is the clear link between an effective and exuberant inflammatory reaction and resolution with tissue repair. The ability of CCL3 application to restore a proper neutrophil response to the infected wounds and as such ensure the appropriate management of the infection is an excellent proof that 'the beginning programs the end' [PMID: 16369558 DOI: 10.1038/ni1276]. On a similar vein, dissecting how resolution starts is equally challenging and interesting [PMID: 27199985 PMCID: PMC4845539 DOI: 10.3389/fimmu.2016.00160] and well presented here. Finally, the notion that neutrophils are not just BAD is also excellent and the study here provides a very clear evidence for the link between onset and resolution [PMID: 27021499 DOI: 10.1016/j.smim.2016.03.007]. This issue may need to be highlighted in the Discussion section as chiefly demonstrated here.I have some comments and some unclear issues I would like the Authors to comment or address.1. FPR1 mRNA (Figure 2)I was surprised to see the rapid decade in FPR1 mRNA within 1 hour of high glucose. To my knowledge, FPR1 is not an early response gene (like fos and jun) so it will be appropriate for the Authors to perhaps double check this dataset, perhaps running a time course and study what else goes down within 30-60-90-120 min. Would there be mileage in monitoring FPR2 mRNA too as a relevant control?

We appreciate the concern raised by the reviewer about the quick reduction in FPR1 at transcriptional and translation level after exposure to HG for 1h. We were also surprised by these results. We have done a follow-up assessment as requested by the reviewer. We confirm our initial data and show that exposure to HG (for 60, 90, and 180 minutes) causes substantial reduction in both FPR1 and FPR2 mRNA (as assessed by RT-PCR) and protein levels (as assessed by Western blotting) (Please see Figure 2—figure supplement 2 and Figure 2—figure supplement 3). We have submitted a grant proposal to determine the mechanism underlying HG-induced repression of FPR expression. We believe that it involves changes in the metabolic state of the neutrophils, caused by exposure to HG.

2. FPR1 proteinThe same question can be asked for the rapid downregulation of FPR1 protein from the cell surface of mouse and human neutrophils following incubation with glucose. The data are convincing here, but again very intriguing how this could occur. More speculation in the Discussion would be good. Also, experiments aiming to assess if the receptor internalised (thus do we find higher intracellular levels?) and if so, how, would be appropriate for a study published in eLife.

We appreciate the reviewer’s astute comments regarding the rapid decline in the surface expression of FPR receptor in neutrophils in response to exposure to HG. We do not know the answer to this but it is possible that it may be merely a reflection of ~90% reduction in the mRNA levels of FPR1, combined with normal FPR1 cellular turnovers. As the reviewer is aware, most cellular receptors are subject to internalization and cycling (62-64). It is also possible that exposure to HG additionally accelerates FPR1 surface protein internalization and/or protein degradation and/or turnovers. More detailed studies are needed to tease out these possibilities and to determine the mechanism underlying HG-induced reduction in FPR surface receptor.

3. Legends to the FiguresI find the figure legends written in an unclear manner. usually the Panel letter (e.g. A, or B or C…) is at the beginning of the sentence, whereas here is all a bit mixed up. In some cases groups of 3 mice have been used. Is this acceptable? How reproducible are the Western blots?

We apologize to the reviewer for the confusions in the figure legends. Our purpose for writing the figure legends the way we did was to reduce the length of the figure legends. For example, if 5 out of 7 panels in a figure shared the same procedure for wounding, treatment, and/or infection but differed only on data analyses (i.e., Western blot, RT-PCR, and flow cytometry, etc), we indicated panel letters (a-e) in the beginning prior to providing s brief description of the protocol, followed by specific letters to indicate the type of analyses after. As for the n (numbers used), we had performed these Western blotting experiments two to three independent times, each time in duplicates or we had used 3- 4 mice per group each time. As the reviewer appreciates, the purpose of “n” is to reach enough statistical power to reach significance and the minimum number of trials per group should be at least N=3. We are asked by our IACUC director at Rush to prevent the unnecessary use of animals in our studies where possible. In addition, these animals (particularly db/db mice) are very expensive further necessitating the prudent use of animals in our studies. Of note, these Western blot experiments are very reproducible. Please see the new data in Figure 2—figure supplement 2 and Figure 2—figure supplement 3 in which the impact of HG on FPR1 and FPR2 expression was assessed.

4. CCL3Data in Figure 4C seem to indicate that the infiltrating neutrophils are responsible for delivering CCL3 (or MIP-1alpha). Probably some indication of the potential sources of CCL3 in the infected wounds, including the migrated neutrophils, is worthwhile. And relevant and interesting.

We appreciate the reviewer’s astute comments. We believe several factors may be contributing. We have added the following paragraph in the Discussion section to address the source of the ligands (including CCL3/MIP-1a) for auxiliary receptors in infected wounds.

“Our data demonstrate that at least the expression and signaling through CCR1 and CXCR2 auxiliary receptors are not adversely affected by high glucose, but they may not be signaling in diabetic wounds early after injury because of insufficient production of their ligands, such as CCL3. […] TLR signaling has also been implicated in the production of ligands (e.g., CCL3) for these auxiliary receptors (46, 47).”

Reviewer #3 (Recommendations for the authors):The manuscript by Roy et al. entitled "Overriding defective FPR chemotaxis signaling in diabetic neutrophil stimulates infection control in diabetic wound", focuses on impaired healing in diabetic wounds. Moreover, the role that neutrophils play and the therapeutic potential of the pro-inflammatory cytokine CCL3 in the treatment of diabetic foot ulcers. Overall, this is an interesting manuscript, but there are several points which should be addressed:• The n values for this study seem low, n=2 or 2 in some cases. This reviewer would argue the validity and robustness of performing statistical analysis on such low n numbers and the ultimate value of the findings.

We apologize to the reviewer for this confusion in the figure legends. Nowhere in this manuscript did we rely on N=2 for our analyses and we completely agree with the reviewers that N=2 is not statistically robust. We indicated N=2 for experiments involving RT-PCR but we had repeated these experiments at least two independent times so the actual number is N>4. We have corrected the figure legends accordingly to address the reviewer’s concern.

• In every figure legend and methods section, the authors need to state fully the n number to provide transparency in their data and in line with eLife policies.

We have done as requested by the reviewer.

• Methods section: Neutrophil isolation from human and mouse. "Mouse neutrophils involving comparisons between C57B normal and db/db diabetic neutrophils were extracted from N=4 blood pools/group, with each blood pool being from 4 mice: totaling 16 mice per group." This is incorrect. There are not 16 mice per group. If each group has blood pooled from 4 mice, then this is n=1, not 4, therefore n=4, not 16. This needs to be addressed.

We apologize to the reviewer for indicating N=16 in the animal experiments where 4 blood pools (each blood pool was blood from 4 mice combined together) were used. We agree with the reviewer that in reality N=4 blood pools per group (generated from 16 mice). We have made the corrections, regarding our statistical analyses both in the figure legends and in the Methods section of the revised manuscript.

• Methods section: Histological analyses and wound healing assessment. "The histological data, (obtained from n≥5 mice/group and >9 random fields/wound/mouse), were normalized per wound surface area". Although the authors state ≥5 mice/group, the figure legends suggest differently. This needs to be corrected.

We have revised the legend to clarify this issue. In these analyses, the number of mice per group was indeed N>5. However, we had analyzed at least 9 random fields per wound per mouse.

• What are the baseline cells counts (especially neutrophils) for the mice used in the study. These should be included.

We appreciate the reviewer’s comment in this regard, but we respectfully disagree with the reviewer that the baseline numbers are necessary for the following reasons. Neutrophils are rarely found in skin and are not considered skin-resident cells, but they migrate into the skin in high numbers in response to inflammatory conditions, such as injury and infection (35-39). We have also demonstrated that in the absence of infection and injury, we do not detect neutrophils in the skin of mice (37). For these reasons, we left out the baseline neutrophil numbers at T0, and only focused on day 1, 3, 6, and 10 wounds to assess the dynamics of neutrophil responses during the acute (early) and chronic (late) stages of healing in in normal and diabetic wounds. In order to produce the baseline numbers, we would need to repeat these studies as we did not maintain day 0 skin wounds from these studies. As the reviewer appreciates, diabetic animals are very expensive and in lieu of the reasons we discussed above, we do not believe that these data are necessary or add anything new to the manuscript.

• In some cases, the authors use bone marrow derived neutrophils and in other cases blood borne neutrophils, although it is unclear to this reviewer what determined the choice. As it is known that these compartments are very different, it would add value to the manuscript to show whether neutrophils derived from these two compartments behave differently and express different cell surface receptors. Furthermore, these changes/similarities should be shown over the time course of 1, 3, 6 and 10 days, as this may help to further explain what is causing the delay in neutrophil influx in diabetic wounds over time.

We appreciate and agree with the reviewer that neutrophils extracted from bone marrow, and blood are not the same. However, our decision to use neutrophils from these different sources was deliberate because we intended to address different questions using these neutrophils. The initial observation (showing impaired neutrophil chemotaxis in diabetic human) involved neutrophils that were isolated from the blood of diabetic patients (10). Therefore, we used isolated neutrophils from the blood of diabetic mice to show that the same chemotaxis impairment was also present in neutrophils isolated from the blood of diabetic mice, as a way to reproduce the human diabetic condition with respect to this impairment. On the other hand, when we wished to assess the impact of HG on chemotaxis and chemotaxis signaling in neutrophils, we isolated neutrophils from the bone marrow of normal mice and exposed them to HG to show that exposure to HG was sufficient to cause similar chemotaxis impairment in normal neutrophils. We used bone marrow neutrophils to remove any potential confounding factors that could potentially influence neutrophil’s chemotaxis behavior. As the reviewers and the editor are aware, neutrophils are produced in bone marrow and released into circulation (30, 31), thus the choice of bone marrow as the source of normal neutrophils. Of note, bone marrow leukocytes (including neutrophils) in diabetic animals do not appear to have the same functional impairments as peripheral blood leukocytes in diabetic animals or human (11, 32-34), suggesting that leukocytes in the bone marrow of diabetic mice are protected from the adverse effects of exposure to HG in the bone marrow compartment. Finally, it was also more practical to use bone marrow neutrophils because we could obtain more neutrophils from this compartment. As we had indicated in our manuscript, we had to pool the blood of 4 mice (16 mice/group) to be able to obtain enough neutrophils for a single analysis.

• Although CCL3 expression is reduced in day 1 diabetic wounds, what happens at 3, 6 and 10 days?

Although we agree with the reviewer that it would be interesting to evaluate the expression of CCL3 at later timepoints, we respectfully believe that this information is not necessary and could possibly detract from the main point of this manuscript for the following reason. The point of this manuscript was why diabetic neutrophils were not trafficking into wounds early after injury if their auxiliary receptors (i.e., CCR1 and CXCR2) remain functional under diabetic condition. This was the reason for us evaluating and showing that CCL3 expression (ligand for CCR1) was significantly diminished in diabetic wounds early after injury. CCL3 is one of many pro-inflammatory cytokines that can recruit neutrophils by engaging ~30 auxiliary receptors (40, 42, 55-59). We used CCL3 to jumpstart the neutrophil response in diabetic wound early after injury because it engages multiple auxiliary receptors on neutrophils, namely CCR1, CCR4, and CCR5 (13-15). As our data show, early after injury diabetic wound environment suffers from inadequate neutrophil response whereas diabetic wounds become progressively more inflamed and contain more neutrophils as they age (Figure 1) and (65). If we assess CCL3 and find it to be elevated in old diabetic wounds (which we think will be the case), then it would raise the question as to what causes CCL3 expression to be elevated in old wounds. On the other hand, if CCL3 expression remains low in the old wounds, it would raise another question as to what is responsible for recruiting neutrophils into old diabetic wounds. We are very much interested in determining the molecular causes responsible for transitioning diabetic wounds from anti-inflammatory (early after injury) into pro-inflammatory (old wounds), but we believe that more comprehensive studies are needed to assess diabetic wound environments during acute and chronic phases of healing and determine and validate the factors (e.g., CCL3 and other pro or anti-inflammatory cytokines) that may be regulated or regulate this transition in diabetic wounds. As the reviewer appreciates such assessments go beyond the scope of this manuscript.

• The finding that topical treatment enhances neutrophil response (figure 4) is very interesting. These data are for day 1, what happens over the time course i.e. 3, 6 and 10 days?

We have done these studies (Please see Figure 5 in the revised manuscript). Our data indicate that as compared to mock-treated diabetic wounds, CCL3-treated diabetic wounds contain significantly more neutrophils at day 1 and 3 but significantly fewer neutrophils at day 6 and day 10, indicating that diabetic wounds are not destined to develop persistent non-resolving inflammation, provided that we jumpstart the neutrophil responses in them early after injury.

• The authors show nicely the involvement of FPR1, but what about the other known FPR that neutrophils express i.e. FPR2? Does this have a compensatory role or is this receptor also dysfunctional. Data should be provided.

We appreciate the reviewers astute comments regarding FPR2. We had originally focused on FPR1 in this manuscript because FPR1 has a much higher affinity for bacterial formyl peptides than FPR2, whereas FPR2 has a broader range of ligands than FPR1 (1, 2). In addition, FPR1 has been implicated in pro-inflammatory responses whereas FPR2 has also been implicated in resolution of inflammation in response to Annexin A1, lipoxin A4, and resolving D1 inflammation pro-resolving agonists (1-5), and because this manuscript was about diabetic neutrophil’s impaired chemotactic response toward infection during the pro-inflammatory phase of acute infection and healing. With that said, we now provide new data (please see Figure 2—figure supplement 2 and Figure 2—figure supplement 3) to show that the expression of both FPR1 and FPR2 are substantially dampened in neutrophils after exposure to HG for 1, 2, and 3 hours.

• In addition, what happens to FPR1 (and 2) expression over the time course i.e. 3, 6 and 10 days? Does it change?

As the reviewer appreciates and as we discussed above, we believe that expanding neutrophil characterizations with respect to FPR and auxiliary receptors and their respective ligands beyond day 1 would go beyond the scope of this manuscript. The point of this manuscript was the novel finding that inadequate neutrophil response early after injury is rendering diabetic wound vulnerable to infection. We believe a comprehensive assessments of all 30 chemotaxis receptors (i.e., FPR1, FPR2, CCR1, CCR4, CCR5, CXCR1, CXCR2, etc.) on neutrophils and their putative ligands (CCL3, CCL5, CXCL1, CXCL2, etc) in normal and diabetic wounds would be needed to shed light on the dynamics of neutrophil responses in normal wound and impaired neutrophil responses in diabetic wounds. In fact, we have submitted a grant proposal to do these studies.

• Were the mice male or female or a mixture? Please provide rational for choice and whether sex as a biological variable was considered, especially given the fact that gender differences have been shown in recovery to diabetic wounds.

We appreciate and agree with the reviewer that sex is a critical biological variable that needs to be considered in designing any experiment. However, since several prior publications have indicated that type 2 diabetic mellitus is more common in males than females [100-103], and because of the budgetary constraints, we restricted our analysis to male db/db (type 2 obese) diabetic mice. Of note, in studies involving human neutrophils, neutrophils were isolated from both male and female volunteers, and regardless of their sex, these neutrophils behaved similarly with respect to their chemotactic responses toward fMLF and CCL3 and the adverse impact of HG on FPR expression and signaling, suggesting that sex does not affect neutrophil chemotaxis behavior or signaling at least in cell culture. Of course, we acknowledge that in vivo, the situation may be different. Therefore, a follow-up study to evaluate the impact of sex on neutrophil’s behavior and function is needed to address this important question.

• The authors chose mice at 8 weeks, with 1 week acclimatisation. Given this young adult age, what is the clinical relevance of this choice of age?

We had previously shown that db/db mice are diabetic at this age (as manifested by their high serum glucose) (54, 65). Unfortunately, the serum glucose levels decline in db/db animals and theses mice short life span because these mice develop ketosis after a few months (66-68). Therefore, we chose this age. We appreciate that type 2 diabetes is more common in middle-aged and elderly people whereas, type 1 diabetes is more common in younger people. However, the rate of type 2 diabetes in young people in USA is increasing substantially due to increasing obesity rates ((69) and according to the Center for Disease Control and Prevention), which db/db mice model at this age represents.

• Figure S3: 100nM fMLP: The data for the 90 mg/dl is around 800 RFU, however, in Figure 2f, this value looks to be much higher (around 1200 RFU). Would the authors provide the exact numbers and discuss further.

As the reviewer appreciates, many factors can cause intrinsic variations in data, including the operator and/or source of reagents. In our experience, even using the fluorescent dye from the same vial and at same concentration can sometimes lead to different intensities and RFU numbers in replicate experiments, particularly if the experiments are performed by different individuals, which was the case for Figure 2f and Figure S3. The important point is that in both experiments, we did obtain similar significant trends as neutrophils treated with normal glucose level (90 mg/dl) displayed significantly higher chemotaxis as compared to neutrophils treated with HG. As requested by the reviewer, we provide the raw data and the exact RFUs for Figure 2f (100nM fMLF) and Figure 2—figure supplement 1 (at indicated fMLF concentrations).

• Although this student specifically looks at neutrophils, do the authors have any data to suggest whether these finds are neutrophil specific or whether e.g. monocytes, play a role in the later time points?

We had focused on neutrophils because the original observation involved chemotaxis impairment in neutrophils isolated from diabetic patients (10) and because *Pseudomonas aeruginosa* infection control in CCL3-treated diabetic wound was primarily dependent on neutrophils as our data show (Figure 4h-j). We appreciate the reviewer’s comment and agree with the reviewer that it would be interesting to assess whether the same impairment in signaling through FPR also affects other leukocytes. We believe it does and have submitted a grant proposal to expand our findings to other leukocytes. However, as the reviewer appreciates, these studies go beyond the scope of this manuscript.

• C57/B should be written in full: C57Bl/6. 'FPR receptor' should be changed to just 'FPR'. Proof reading would help – grammar and spelling mistakes need correcting.

We thank the reviewer for his/her suggestions and have made the corrections in the revised manuscript.

References:

1. D. Y. Richard et al., International Union of Basic and Clinical Pharmacology. LXXIII. Nomenclature for the formyl peptide receptor (FPR) family. Pharmacological reviews 61, 119-161 (2009).

2. Y. S. Jeong, Y.-S. Bae, Formyl peptide receptors in the mucosal immune system. Experimental and molecular medicine 52, 1694-1704 (2020).

3. S. Bena, V. Brancaleone, J. M. Wang, M. Perretti, R. J. Flower, Annexin A1 interaction with the FPR2/ALX receptor: identification of distinct domains and downstream associated signaling. Journal of Biological Chemistry 287, 24690-24697 (2012).

4. S. Yazid, L. V. Norling, R. J. Flower, Anti-inflammatory drugs, eicosanoids and the annexin A1/FPR2 anti-inflammatory system. Prostaglandins and other lipid mediators 98, 94-100 (2012).

5. C. N. Serhan, J. Savill, Resolution of inflammation: the beginning programs the end. Nature immunology 6, 1191-1197 (2005).

6. L. R. Kalan, M. B. Brennan, The role of the microbiome in nonhealing diabetic wounds. Annals of the New York Academy of Sciences 1435, 79-92 (2019).

7. E. Cengiz, W. V. Tamborlane, A tale of two compartments: interstitial versus blood glucose monitoring. Diabetes Technol Ther 11 Suppl 1, S11-16 (2009).

8. L. S. Barcelos et al., Human CD133+ progenitor cells promote the healing of diabetic ischemic ulcers by paracrine stimulation of angiogenesis and activation of Wnt signaling. Circ Res 104, 1095-1102 (2009).

9. D. G. Armstrong, A. J. Boulton, S. A. Bus, Diabetic foot ulcers and their recurrence. New England Journal of Medicine 376, 2367-2375 (2017).

10. M. Delamaire et al., Impaired leucocyte functions in diabetic patients. Diabetic Medicine 14, 29-34 (1997).

11. J. E. Repine, C. C. Clawson, F. C. Goetz, Bactericidal function of neutrophils from patients with acute bacterial infections and from diabetics. The Journal of infectious diseases 142, 869-875 (1980).

12. S. Gallacher et al., Neutrophil bactericidal function in diabetes mellitus: evidence for association with blood glucose control. Diabetic medicine 12, 916-920 (1995).

13. C. D. Ramos et al., MIP-1alpha[CCL3] acting on the CCR1 receptor mediates neutrophil migration in immune inflammation via sequential release of TNF-α and LTB4. Journal of leukocyte biology 78, 167-177 (2005).

14. J. M. da Silva et al., Relevance of CCL3/CCR5 axis in oral carcinogenesis. Oncotarget 8, 51024 (2017).

15. O. Yoshie, K. Matsushima, CCR4 and its ligands: from bench to bedside. International immunology 27, 11-20 (2015).

16. M. K. Nagai, J. M. Embil, Becaplermin: recombinant platelet derived growth factor, a new treatment for healing diabetic foot ulcers. Expert opinion on biological therapy 2, 211-218 (2002).

17. D. G. Greenhalgh, K. H. Sprugel, M. J. Murray, R. Ross, PDGF and FGF stimulate wound healing in the genetically diabetic mouse. Am J Pathol 136, 1235-1246 (1990).

18. R. Tsuboi, D. B. Rifkin, Recombinant basic fibroblast growth factor stimulates wound healing in healing-impaired db/db mice. The Journal of experimental medicine 172, 245-251 (1990).

19. K. Obara et al., Photocrosslinkable chitosan hydrogel containing fibroblast growth factor-2 stimulates wound healing in healing-impaired db/db mice. Biomaterials 24, 3437-3444 (2003).

20. G. L. Brown et al., Enhancement of wound healing by topical treatment with epidermal growth factor. The New England journal of medicine 321, 76-79 (1989).

21. J. M. Smiell, Clinical safety of becaplermin (rhPDGF-BB) gel. Becaplermin Studies Group. American journal of surgery 176, 68S-73S (1998).

22. R. S. Howell-Jones et al., A review of the microbiology, antibiotic usage and resistance in chronic skin wounds. J Antimicrob Chemother 55, 143-149 (2005).

23. M. Abbas, I. Uckay, B. A. Lipsky, In diabetic foot infections antibiotics are to treat infection, not to heal wounds. Expert Opin Pharmacother 16, 821-832 (2015).

24. S. Becattini, Y. Taur, E. G. Pamer, Antibiotic-Induced Changes in the Intestinal Microbiota and Disease. Trends Mol Med 22, 458-478 (2016).

25. H. E. Jakobsson et al., Short-term antibiotic treatment has differing long-term impacts on the human throat and gut microbiome. PloS one 5, e9836 (2010).

26. A. Langdon, N. Crook, G. Dantas, The effects of antibiotics on the microbiome throughout development and alternative approaches for therapeutic modulation. Genome medicine 8, 39 (2016).

27. K. Korpela et al., Intestinal microbiome is related to lifetime antibiotic use in Finnish pre-school children. Nat Commun 7, 10410 (2016).

28. M. Yassour et al., Natural history of the infant gut microbiome and impact of antibiotic treatment on bacterial strain diversity and stability. Sci Transl Med 8, 343ra381 (2016).

29. T. R. Sampson et al., Gut Microbiota Regulate Motor Deficits and Neuroinflammation in a Model of Parkinsons Disease. Cell 167, 1469-1480.e1412 (2014).

30. R. C. Furze, S. M. Rankin, Neutrophil mobilization and clearance in the bone marrow. Immunology 125, 281-288 (2008).

31. P. C. Burdon, C. Martin, S. M. Rankin, The CXC chemokine MIP-2 stimulates neutrophil mobilization from the rat bone marrow in a CD49d-dependent manner. Blood 105, 2543-2548 (2005).

32. I. L. Scully et al., Neutrophil killing of *Staphylococcus aureus* in diabetes, obesity and metabolic syndrome: a prospective cellular surveillance study. Diabetology and metabolic syndrome 9, 76 (2017).

33. A. A. Sima, S. J. O'Neill, D. Naimark, S. Yagihashi, D. Klass, Bacterial phagocytosis and intracellular killing by alveolar macrophages in BB rats. Diabetes 37, 544-549 (1988).

34. S. Park, J. Rich, F. Hanses, J. C. Lee, Defects in innate immunity predispose C57BL/6J-Leprdb/Leprdb mice to infection by *Staphylococcus aureus*. Infection and immunity 77, 1008-1014 (2009).

35. A. V. Nguyen, A. M. Soulika, The Dynamics of the Skin's Immune System. Int J Mol Sci 20, (2019).

36. K. Passelli, O. Billion, F. Tacchini-Cottier, The Impact of Neutrophil Recruitment to the Skin on the Pathology Induced by Leishmania Infection. Front Immunol 12, 649348 (2021).

37. J. L. Hamilton, M. F. Mohamed, B. R. Witt, M. A. Wimmer, S. H. Shafikhani, Therapeutic assessment of N-formyl-methionyl-leucyl-phenylalanine (fMLP) in reducing periprosthetic joint infection. Eur Cell Mater 41, 122-138 (2021).

38. T. A. Wilgus, S. Roy, J. C. McDaniel, Neutrophils and Wound Repair: Positive Actions and Negative Reactions. Adv Wound Care (New Rochelle) 2, 379-388 (2013).

39. S. de Oliveira, E. E. Rosowski, A. Huttenlocher, Neutrophil migration in infection and wound repair: going forward in reverse. Nat Rev Immunol 16, 378-391 (2016).

40. R. C. Chou et al., Lipid-cytokine-chemokine cascade drives neutrophil recruitment in a murine model of inflammatory arthritis. Immunity 33, 266-278 (2010).

41. Y. Su, A. Richmond, Chemokine regulation of neutrophil infiltration of skin wounds. Advances in wound care 4, 631-640 (2015).

42. P. V. Afonso et al., LTB4 is a signal-relay molecule during neutrophil chemotaxis. Developmental cell 22, 1079-1091 (2012).

43. A. D. Luster, R. Alon, U. H. von Andrian, Immune cell migration in inflammation: present and future therapeutic targets. Nature immunology 6, 1182-1190 (2005).

44. A. Ridiandries, J. T. M. Tan, C. A. Bursill, The Role of Chemokines in Wound Healing. Int J Mol Sci 19, (2018).

45. R. Roy et al., IL-10 Dysregulation Underlies Chemokine Insufficiency, Delayed Macrophage Response, and Impaired Healing in Diabetic Wound. Journal of Investigative Dermatology, (2021).

46. S. Kochumon et al., Stearic Acid and TNF-α Co-Operatively Potentiate MIP-1alpha Production in Monocytic Cells via MyD88 Independent TLR4/TBK/IRF3 Signaling Pathway. Biomedicines 8, (2020).

47. R. Ahmad et al., MIP-1alpha induction by palmitate in the human monocytic cells implicates TLR4 signaling mechanism. Cellular physiology and biochemistry : international journal of experimental cellular physiology, biochemistry, and pharmacology 52, 212-224 (2019).

48. I. Bhavsar, C. S. Miller, M. Al-Sabbagh, Macrophage inflammatory protein-1 α (MIP-1 α)/CCL3: as a biomarker. General methods in biomarker research and their applications, 223–249 (2015).

49. H. Q. He et al., Functional characterization of three mouse formyl peptide receptors. Molecular pharmacology 83, 389-398 (2013).

50. A. Zibert et al., CCL3/MIP-1 α Is a Potent Immunostimulator When Coexpressed with Interleukin-2 or Granulocyte-Macrophage Colony-Stimulating Factor in a Leukemia/Lymphoma Vaccine. Human gene therapy 15, 21-34 (2004).

51. H. R. Jones, C. T. Robb, M. Perretti, A. G. Rossi, The role of neutrophils in inflammation resolution. Semin Immunol 28, 137-145 (2016).

52. C. Wetzler, H. Kampfer, B. Stallmeyer, J. Pfeilschifter, S. Frank, Large and sustained induction of chemokines during impaired wound healing in the genetically diabetic mouse: prolonged persistence of neutrophils and macrophages during the late phase of repair. The Journal of investigative dermatology 115, 245-253 (2000).

53. T. Bjarnsholt et al., Why chronic wounds will not heal: a novel hypothesis. Wound repair and regeneration: official publication of the Wound Healing Society [and] the European Tissue Repair Society 16, 2-10 (2008).

54. J. Goldufsky et al., *Pseudomonas aeruginosa* uses T3SS to inhibit diabetic wound healing. Wound repair and regeneration: official publication of the Wound Healing Society [and] the European Tissue Repair Society 23, 557-564 (2015).

55. S. de Oliveira, E. E. Rosowski, A. Huttenlocher, Neutrophil migration in infection and wound repair: going forward in reverse. Nature reviews. Immunology 16, 378 (2016).

56. M. Liu et al., Formylpeptide receptors are critical for rapid neutrophil mobilization in host defense against *ListeriaListeria monocytogenes*. Scientific reports 2, 786 (2012).

57. C. D. Sadik, N. D. Kim, A. D. Luster, Neutrophils cascading their way to inflammation. Trends in immunology 32, 452-460 (2011).

58. K. Futosi, S. Fodor, A. Mocsai, Neutrophil cell surface receptors and their intracellular signal transduction pathways. Int Immunopharmacol 17, 638-650 (2013).

59. L. G. Ng et al., Visualizing the neutrophil response to sterile tissue injury in mouse dermis reveals a three-phase cascade of events. Journal of Investigative Dermatology 131, 2058-2068 (2011).

60. S. Oldekamp et al., Lack of formyl peptide receptor 1 and 2 leads to more severe inflammation and higher mortality in mice with of pneumococcal meningitis. Immunology 143, 447-461 (2014).

61. J. L. Gao, E. J. Lee, P. M. Murphy, Impaired antibacterial host defense in mice lacking the N-formylpeptide receptor. The Journal of experimental medicine 189, 657-662 (1999).

62. Y. Minami, L. E. Samelson, R. D. Klausner, Internalization and cycling of the T cell antigen receptor. Role of protein kinase C. The Journal of biological chemistry 262, 13342-13347 (1987).

63. S. Tomita, M. Fukata, R. A. Nicoll, D. S. Bredt, Dynamic interaction of stargazin-like TARPs with cycling AMPA receptors at synapses. Science 303, 1508-1511 (2004).

64. J. Chen, J. Wang, K. R. Meyers, C. A. Enns, Transferrin-directed internalization and cycling of transferrin receptor 2. Traffic 10, 1488-1501 (2009).

65. S. Wood et al., Pro-inflammatory chemokine CCL2 (MCP-1) promotes healing in diabetic wounds by restoring the macrophage response. PloS one 9, e91574 (2014).

66. K. P. Hummel, M. M. Dickie, D. L. Coleman, Diabetes, a new mutation in the mouse. Science 153, 1127-1128 (1966).

67. K. Srinivasan, P. Ramarao, Animal models in type 2 diabetes research: an overview. The Indian journal of medical research 125, 451-472 (2007).

68. A. J. King, The use of animal models in diabetes research. Br J Pharmacol 166, 877-894 (2012).

69. T. A. Hillier, K. L. Pedula, Complications in young adults with early-onset type 2 diabetes: losing the relative protection of youth. Diabetes care 26, 2999-3005 (2003).